# Zero-shot Human Pose Estimation using Diffusion-based Inverse solvers

**Sahil Bhandary Karnoor & Romit Roy Choudhury**
University of Illinois at Urbana-Champaign
Champaign, IL 61820, USA
{sahilb5,croy}@illinois.edu

## Abstract

Pose estimation refers to tracking a human's full body posture, including their head, torso, arms, and legs. The problem is challenging in practical settings where the number of body sensors is limited. Past work has shown promising results using conditional diffusion models, where the pose prediction is conditioned on both ⟨location, rotation⟩ measurements from the sensors. Unfortunately, nearly all these approaches generalize poorly across users, primarily because location measurements are highly influenced by the body shape of the user. In this paper, we formulate pose estimation as an inverse problem and design an algorithm capable of zero-shot generalization. Our idea utilizes a pre-trained diffusion model and conditions it on rotational measurements alone; the priors from this model are then guided by a likelihood term, derived from the measured locations. Thus, given any user, our proposed `InPose` method generatively estimates the highly likely sequence of poses that best explains the sparse on-body measurements.

## 1 Introduction

Human pose estimation is a crucial piece for numerous applications, The results have steadily improved Dittadi et al. (2021); Pavlakos et al. (2019) with a recent boost from generative models (e.g., conditional diffusion models Castillo et al. (2023)) that predicted the user's full-body pose from just 3 sensors. Unfortunately, such proposed generative techniques have a significant limitation; they don't generalize well across users with varying body shapes. A generative model trained on data from a single user can't be used by a user with a different body shape without fine-tuning. Authors in Aliakbarian et al. (2022) try to overcome this issue by jointly training over both pose datasets and varying body shapes, but this increases model complexity, and there is no guarantee that all possible body shapes were accounted for during training. An algorithm that generalizes even to body shape outliers would be ideal.

In this paper, we propose `InPose`, a diffusion-based method that implicitly accounts for the user's body shape without requiring any fine-tuning. Our core observation is that any human's full-body pose can be decomposed into a "scale-free pose" and a "scale-dependent" component. For human poses, the scale-free pose can be imagined as a template human body whose skeletal joints (e.g., shoulders, elbows, hip, knees, etc.) are rotated appropriately to create a given pose. The scale-dependent component is the joint locations in 3D space. Forward kinematics relates the scale-free pose, along with the body shape, to the scale-dependent component. Since the sensors give ⟨rotation, location⟩ measurements from 3 body joints, it is possible to estimate a distribution of scale-free poses from rotational measurements alone. Then, the location measurements can be used to sharpen this distribution to poses that best explain the measurements. This decomposition lends itself to an inverse problem formulation, shown visually in Fig. 1a. Using ⟨rotation, location⟩ measurements from 3 body joints—head and two wrists—`InPose` aims to track the locations of all 22 body joints, necessary to fully define the 3D pose of a human.

`InPose`'s inverse problem formulation can be sketched as follows. We train a Diffusion model conditioned on *rotational measurements* from existing datasets; this gives us a conditional prior over scale-free poses. When inferring a specific user's pose, we use the user's body shape to scale the scale-free pose, and compare it against *location measurements* to estimate the likelihood of the pose.

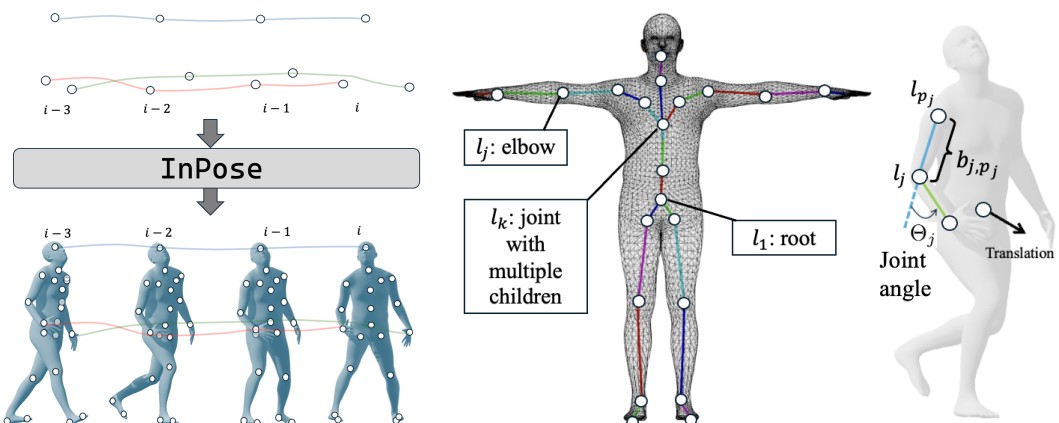

Figure 1: (a) `InPose`'s input and output visualized over 4 time frames. (b) "T" pose. (c) Pose with a depiction of the rotation angle and root translation.

This likelihood term requires propagating a Gaussian random variable through a nonlinear operator. We prove that this propagation can be approximated by a Gaussian and use the likelihood as an inverse kinematics guidance term to guide the diffusion denoising process. The denoised result is a sequence of full-body poses—samples from the posterior—that best explains the 3-point measurements for that specific user. Through extensive experiments, we demonstrate promising generalization across a wide range of body shapes and shapes on the AMASS dataset Mahmood et al. (2019).

## 2  MODEL AND MEASUREMENT

**Body Model:** Following the conventional SMPL framework Loper et al. (2015), we model the human body as a graph (Fig. 1b). The vertices of this graph are the 22 main joints in the human skeleton; the edges are the bones connecting these joints. The 3D coordinates of the joints (in a global reference frame) are denoted as $l_j \in \mathbb{R}^3, j \in \{1, ..22\}$. The bone that connects adjacent joints $l_j, l_k$ are denoted by a vector $b_{j,k} \in \mathbb{R}^3$ of fixed length $|b_{jk}|$. Every joint $l_j$ has a unique parent, $l_{p_j}$. The whole joint-tree has a root joint $l_1$ located at the pelvis.

The *global pose* of a body is fully defined by the 22 joint locations in a global reference frame. Fig.1b shows a "T" pose and Fig.1c shows a running pose. Intuitively, a global pose can be computed in three steps. (1) Start with a standard "T" pose with the human located at the origin of a global reference frame. (2) Move the root joint $l_1$ to bring the human to its correct location; the human is still in the "T" pose, but the whole body is displaced. (3) Now, starting from joint $l_1$, rotate each joint based on the joint angles. Perform this *sequentially down the joint tree*, ensuring a parent joint $p_j$ has been rotated before rotating joint $j$. These 3 steps bring us to the human's global pose.

Eq. 1 models the above steps to compute joint $j$'s global location at time frame $i$.

$$l_j(i) = l_{p_j}(i) + R_{p_j}(i) \cdot b_{j,p_j} \tag{1}$$

Here $R_{p_j}(i)$ is a global 3D rotation matrix of the parent joint. Note that the global 3D rotation matrix for any joint $j$ is computed as $R_j(i) = R_{p_j}(i) \cdot \Theta_j(i)$, where $\Theta_j(i)$ is the *local* 3D rotation matrix shown in Fig.1c. Since $\Theta_j(i)$ is represented as 3D rotation matrices[1], all the joint angles in the "T" pose are identity matrices. As the human performs different poses, `InPose` aims to track the root location $l_1(i)$ and global rotation $R_j(i)$ for each of the joints.

**Joint Angle Representation:** While representing $R_j(i)$ using 3D rotation matrices makes it easy to compute joint locations, Zhou et al. (2019) has shown that representing them instead using the 6D parameterization $r_j(i) \in \mathbb{R}^6$ is better for neural network training. This is due to the continuity properties of this representation, unlike others such as quaternions or axis-angle[2]. The forward

---

[1]Other representations are possible, including axis-angle, Euler, or quaternion representation.

[2]Rotation matrices may also be used, but since they must be made unitary, output pose quality is affected.

mapping for vector $r_j(i)$ is computed as:

$$r_j(i) = [R_j^{(1,1)}(i) \quad R_j^{(2,1)}(i) \quad R_j^{(3,1)}(i) \quad R_j^{(1,2)}(i) \quad R_j^{(2,2)}(i) \quad R_j^{(3,2)}(i)]^\top$$

where $R_j^{(k,l)}(i)$ is the $\{k,l\}^{\text{th}}$ element of the corresponding 3D rotation matrix. There also exists a non-linear differentiable inverse $\bar{\mathcal{D}} : \mathbb{R}^6 \to \mathbb{R}^{3\times3}$ that maps the 6D representation to rotation matrices (defined in Appendix A). Hence, Eq.1 becomes: $l_j(i) = l_{p_j}(i) + \bar{\mathcal{D}}(r_{p_j}(i)) \cdot b_{j,p_j}$. We extend $\bar{\mathcal{D}}$ to map all $|M|$ rotations from 6D to rotation matrices, and term this function $\mathcal{D}$.

**Measurements:** To align with recent work in this area Zheng et al. (2023); Castillo et al. (2023), we use the same AMASS dataset, which contains locations and rotation angles of the head and two wrists. These measurements can be obtained from VR goggles and handheld controllers, which use a combination of ego-centric cameras, IMU sensors, visual SLAM algorithms, and dead reckoning methods to estimate $m$ locations and rotations in the global reference frame, where $m \subset M = \{1, ...22\}$ and $|m| = 3$.

## 3 INPOSE: INVERSE ZERO-SHOT POSE ESTIMATION

### 3.1 FORMULATION

We denote the noisy signal measurements from the 3 sensor joints as $y_m(i) = [l_m(i), r_m(i)]$. Here $i$ indexes the measurement time frames and is dropped throughout the rest of the paper unless specified. $l_m(i) = l_m^+(i) + \sigma_l v(i)$ is the noisy location measurement, where $l_m^+$ is the noise-free joint location, and $v(i)$ is iid Gaussian noise. Similarly, the rotation $r_m(i)$ is also noisy. Our goal is to predict $r_M$, which are the 22 global joint rotations and $l_1$, the root's translation. We are also provided the user's bone lengths $b_{j,p_j}$. With such sparse measurements, this is an ill-posed pose estimation problem.

As in Jiang et al. (2022), we simplify this problem by first assuming the root joint is stationary. We estimate the scale-free pose defined by $r_M$; then scale to the correct pose defined by $l_M$; and then *drag* this pose until the head's location matches the measured head location, $l_{\text{head}}$. From this, we infer the root translation $l_1$. Thus, the core question boils down to sampling from the posterior $p(r_M|y_m)$.

Diffusion models have recently found remarkable success for these types of posterior sampling problems. They were originally proposed as a tool for sampling from a prior distribution $p_0(x^0)$. This is done by first defining a noising process $p_t(x^t)$ by injecting iid Gaussian noise of standard deviation $\sigma_t$ into it, where $t \in \{0 : T\}$. Diffusion models aim to reverse this noising process by learning the score function $\nabla_{x_t} log\, p_t(x_t)$.

In our scenario, we require the conditional score $\nabla_{r_M^t} log\, p_t(r_M^t|y_m)$. One method is to use Classifier-Free Guidance (CFG) proposed by Ho & Salimans (2021). In this formulation, a conditional diffusion model is trained to accept all the inputs $y_m = [l_m, r_m]$ for conditioning. Most previous works Castillo et al. (2023); Van Wouwe et al. (2024) use this approach and are unable to support zero-shot generalization. This is because the (noisy) location measurements $l_m$ vary based on body shape—if two people are in the same pose, their joints would share identical rotation angles, *but because of differences in bone lengths across varying body shapes, we see from Eq. 1 that the joint locations $l_j$ will be different.* Thus, a conditional model trained on one user's data does not generalize well to another.

To overcome this, we split the conditional score $\nabla_{r_M^t} log\, p_t(r_M^t|y_m) = \nabla_{r_M^t} log\, p_t(r_M^t|\{l_m, r_m\})$ using Bayes' rule:

$$\nabla_{r_M^t} log\, p_t(r_M^t|\{l_m, r_m\}) = \nabla_{r_M^t} log\, p_t(r_M^t|r_m) + \nabla_{r_M^t} log\, p_t(l_m|r_M^t, r_m) + 0 \qquad (2)$$

$$= \nabla_{r_M^t} log\, p_t(r_M^t|r_m) + \nabla_{r_M^t} log\, p_t(l_m|r_M^t) \qquad (3)$$

where we have assumed $l_m$ and $r_m$ are **conditionally independent**.

The conditional score $\nabla_{r_M^t} log\, p_t(r_M^t|r_m)$ is scale-free, and can be learned by a CFG-based conditional diffusion model. The scale-dependent likelihood score $\nabla_{r_M^t} log\, p_t(l_m|r_M^t)$ can be utilized as a guidance to the prior Chung et al. (2023); Kawar et al. (2022); Rozet et al. (2024); Yang et al. (2024). This guidance is performed during inference and *does not require any training or fine-tuning of the generative neural network.* We use the Pseudoinverse-Guidance for Diffusion Models ($\Pi$GDM) Song

et al. (2023) framework, but propose a mechanism to propagate a Gaussian random variable through the non-linear inverse $\mathcal{D}(.)$ function inside the likelihood term (discussed soon). This mathematically enables the decomposition of the user's pose into a general scale-free pose (from the prior) and a user-specific scaling factor (captured in the likelihood). Thus, our main contribution over past work—and the key to enabling zero-shot pose-prediction—is to perform CFG using *only* the measured rotations, and use *only* the joint locations as a pseudoinverse guidance to that CFG.

## 3.2 Designing the Likelihood, Prior, and Posterior Terms

We now describe the various score terms used by our algorithm at each diffusion timestep.

**CFG Prior:** We train a CFG-based score model $\epsilon_\theta(r_M^t, t, r_m)$ to determine the conditional score $\nabla_{r_M^t} log\ p_t(r_M^t|r_m)$ as a function of the noisy rotation inputs $r_m$ from the 3-point sensors. This is then used to derive a conditionally denoised estimate $\hat{r}_M^t$ using Tweedie's formula Efron (2011):

$$\hat{r}_M^t = \frac{r_M^t - \sqrt{1 - \bar{\alpha}_t}\epsilon_\theta(r_M^t, t, r_m)}{\sqrt{\bar{\alpha}_t}} \tag{4}$$

We adapt the same DiT Peebles & Xie (2022) transformer architecture used in BoDiffusion Castillo et al. (2023) but modify it appropriately (details in Section 4) since we are only conditioning on rotation $r_m$, while BoDiffusion conditioned on both $r_m$ and $l_m$.

**Likelihood score:** To compute the likelihood score $\nabla_{r_M^t} log\ p_t(l_m|r_M^t)$, we need to relate the joint rotations to joint locations. Mathematically, we aim to minimize the likelihood $||l_m - \mathcal{A} \circ \mathcal{D}(\hat{r}_M^t)||_2$ where $\mathcal{A} \circ \mathcal{D}(\cdot)$ is the *measurement* operator. Recall, $\mathcal{D}(.)$ converts 22 joint rotations from the 6D vectors to rotation matrices, and $\mathcal{A}$ is a linear function that uses these rotation matrices to determine the joint location estimates. Eq. 1 had earlier shown the operation of $\mathcal{A}$ for a single joint, where $R_j(i) = \mathcal{D}(r_j(i))$.

Unfortunately, two issues stand in the way of estimating the likelihood. ❶ Since $r_M^t$ is a noisy estimate of $r_M$, we cannot pass it through the measurement operator to obtain $\hat{l}_m$. To mitigate this, we adopt ideas from ΠGDM Song et al. (2023) to help approximate $r_M^t$ as a Gaussian distribution. ❷ Since $\mathcal{D}(.)$ is a non-linear function, propagating the approximated Gaussian random variable through $\mathcal{D}(.)$ is problematic. We will prove that this propagation can also be approximated by a Gaussian, giving us a pathway to the final solution. Let us briefly review ΠGDM first and then visit the second step in `InPose`.

---

*ΠGDM Recap*: Consider the general problem where we are given observations $z = Ax^0 + \sigma_z n$, where $A$ is the measurement model, $n$ is unit Gaussian noise, and $\sigma_z$ is the noise variance. Say we would like to estimate $x^0$, for which we will guide a diffusion model that is denoising a noisy $x^t$ at each diffusion time step. This guidance needs to use the likelihood score $\nabla_{x^t} \log p_t(z|x^t)$, which needs to be computed through an intermediate step of marginalization over $x^0$ as follows:

$$p_t(z|x^t) = \int p(z|x^0)p_t(x^0|x^t)\mathrm{d}x^0 \tag{5}$$

If $A$ is a linear function and the noise $n$ is Gaussian, then $p(z|x^0) \sim \mathcal{N}(Ax^0, \sigma_z I)$. For the second term $p_t(x^0|x^t)$, ΠGDM proposes to approximate this distribution as $\mathcal{N}(\hat{x}^t, w_t^2 I)$, where the mean comes from a regular diffusion step. Hence, the distribution $p_t(z|x^t)$ and the corresponding likelihood score can both be approximated by a Gaussian as follows:

$$p_t(z|x^t) \approx \mathcal{N}(A\hat{x}^t, w_t^2 AA^\top + \sigma_z I) \tag{6}$$

$$\nabla_{x^t} \log p_t(z|x^t) \approx ((z - A\hat{x}^t)^\top (w_t^2 AA^\top + \sigma_z^2 I)^{-1} A \frac{\partial \hat{x}^t}{\partial x^t})^\top \tag{7}$$

---

Let us now return to `InPose`. We cannot directly apply ΠGDM to our likelihood score $\nabla_{r_M^t} log\ p_t(l_m|r_M^t)$ since our measurement operator contains the $\mathcal{D}(.)$ function. But if $\mathcal{D}(.)$ is ignored—meaning that the rotation matrix $R_M^t$ is somehow available—then the measurement operator in Eq. 1 becomes linear. When $l_1 = 0$, the joint location $l_j$ becomes a matrix-vector product, $C\kappa$,

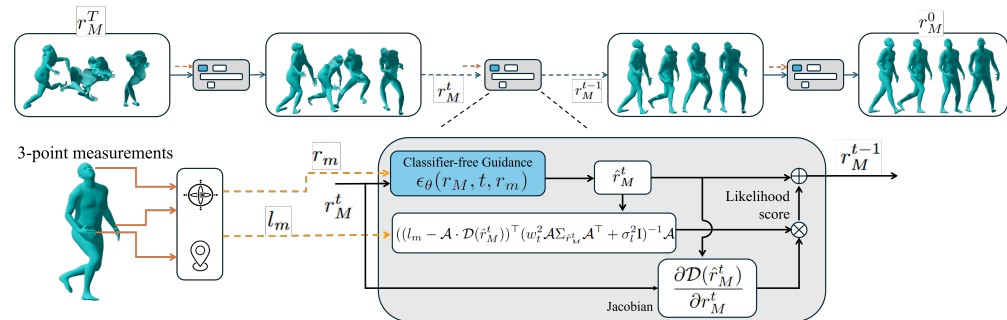

Figure 2: `InPose` pipeline: 3-point sensor rotation + location measurements are inputs. Rotations are fed to the CFG score model, which outputs a conditional prior; location measurements estimate the likelihood, which is used to steer diffusion.

by traversing the joint tree as follows:

$$[R_1...R_{p_j}] \cdot [b_{2,1}^\top...b_{j,p_j}^\top]^\top = l_j \tag{8}$$

where $C = [R_1...R_{p_j}]$, and $\kappa = [b_{2,1}^\top...b_{j,p_j}^\top]^\top$. The matrix-vector product can be rearranged to form $(I_3 \otimes \kappa^\top) \cdot \text{vec}(C)$ where $\otimes$ is the Kronecker product. We can thus obtain our linear function $\mathcal{A} := I_3 \otimes \kappa^\top$. Plugging this $\mathcal{A}$ into Eq 7 gives us the likelihood score.

Unfortunately, the $\mathcal{D}(.)$ function is non-linear in `InPose`, hence the conditional distribution $p_t(z|x^t)$ in Eq. 6 is no longer Gaussian. However, using the following Theorem, we show that it is possible to approximate $p_t(l_m|r_M^t)$ as a Gaussian and compute its covariance matrix with a well-trained score model $\epsilon_\theta$ (proof in Appendix A).

**Theorem 1.** *We are given a well-trained score model $\epsilon_\theta$, that learns the score function $\epsilon_t \leftarrow \epsilon_\theta(r_M^t, t, r_m)$, and denoises $\hat{r}_M^t \leftarrow \frac{r_M^t - \sqrt{1-\bar{\alpha}_t}\epsilon_t}{\sqrt{\bar{\alpha}_t}}$. If the model ensures that $||\hat{r}_j^{t,1:3}|| = ||\hat{r}_j^{t,4:6}|| = 1$, $\langle \hat{r}_j^{t,1:3}, \hat{r}_j^{t,4:6} \rangle = 0$, $\forall j \in M$ then $p_t(\mathcal{D}(r_M^0)|r_M^t) \approx \mathcal{N}(\mathcal{D}(\hat{r}_M^t), w_t^2 \Sigma_{\hat{r}_M^t})$ where $\Sigma_{\hat{r}_M^t}$ is a positive definite matrix.*

From the proof of Theorem 1, we obtain the covariance matrix for $\mathcal{D}(\hat{r}_M^t)$ as $\widetilde{\text{Cov}}(\mathcal{D}(\hat{r}_M^t)) = w_t^2 \Sigma_{\hat{r}_M^t}$. Using this approximation, we get $p_t(\mathcal{D}(r_M^0)|r_M^t) \approx \mathcal{N}(\mathcal{D}(\hat{r}_M^t), w_t^2 \Sigma_{\hat{r}_M^t})$, and thus

$$\nabla_{r_M^t} \log p_t(l_m|r_M^t) = ((l_m - \mathcal{A} \cdot \mathcal{D}(\hat{r}_M^t))^\top (w_t^2 \mathcal{A}\Sigma_{\hat{r}_M^t}\mathcal{A}^\top + \sigma_l^2 I)^{-1} \mathcal{A} \frac{\partial \mathcal{D}(\hat{r}_M^t)}{\partial r_M^t})^\top \tag{9}$$

### 3.3 ACCOUNTING FOR TRANSLATION: DIFFERENTIAL PARAMETERIZATION

From Eq. 1, we see that all joint locations at frame $i$ have an additive dependence on $l_1(i)$ due to the kinematic chain:

$$l_j(i) = \sum_{k=3}^{j}(l_{p_k}(i) + R_{p_k}(i) \cdot b_{k,p_k}) + R_1(i) \cdot b_{2,1} + l_1(i) \tag{10}$$

The mapping $\mathcal{A}$ derived from Eq. 8 is only valid if $l_1(i) = 0$. To enable the linear inverse guidance formulation when $l_1(i) \neq 0$, we use the difference between the positional measurements from each of the 3 measured joints at every $i^{\text{th}}$ frame. Thus, the contribution of the root translation $l_1(i)$ for each of the measured joint locations gets canceled. However, lower body motion estimation remains limited because there is no direct translational guidance. `InPose` relies heavily on the prior learned by the diffusion model to estimate lower body motion.

For body shapes similar to the default shape, we can use the head translation as input to the CFG model, enabling us to estimate lower body motion. We perform experiments both with and without head translation as a CFG-based input.

### 3.4 MODEL PIPELINE

Figure 2 summarizes the `InPose` pipeline. Its objective can be summarized as performing guided diffusion to infer a sequence of human poses $r_M$, using a combination of conditioning CFG inputs

and Pseudoinverse guidance utilizing a modified $\Pi$GDM likelihood score. The inputs to the algorithm are the noisy joint rotations $r_{j\in m}$ and locations $l_{j\in m}$ of a subset of joints $m$ of size 3.

At each diffusion step $t$, `InPose`'s workflow can be summarized in the following steps:

- Use the CFG score function $\nabla_{r_M^t} log\, p_t(r_M^t | r_m)$ conditioned on noisy rotation inputs $\{r_m\}$ ($\epsilon_\theta$) to generate a conditionally denoised estimate $\hat{r}_M^t$.

- Use $\Pi$GDM to estimate the likelihood score $\nabla_{r_M^t} \log p_t(l_m | r_M^t)$.

- Combine the conditionally denoised estimate and the likelihood score using modified DDIM to generate the diffusion output for the next step $r_M^{t-1}$. The proposed `InPose` algorithm is described in Algorithm 1.

---

**Algorithm 1** `InPose` Inference using modified $\Pi$GDM

---

**Require:** $N, \epsilon_\theta, \eta \in [0,1], \mathcal{A}, \mathcal{D}(\cdot)$
   **Inputs:** $y_m = [l_m, r_m], \sigma_l$
   Find a sequence of timesteps $q_{i\in 0..N}$ with $q_0 = 0$ and $q_N = T$
   Initialize $r_M \sim \mathcal{N}(0, \mathrm{I})$
   **for** $i \in \{N..1\}$ **do**
      $t \leftarrow q_i, \quad s \leftarrow q_{i-1}$          $\triangleright$ Get start and end times
      $\bar{\alpha}_t \leftarrow \frac{1}{1+\sigma_t^2}$          $\triangleright$ Get $\alpha$ for VP-SDE

      $\epsilon_t \leftarrow \epsilon_\theta(r_M, t, r_m)$
      $\hat{r}_M^t = \frac{r_M - \sqrt{1-\bar{\alpha}_t}\epsilon_t}{\sqrt{\bar{\alpha}_t}}$          $\triangleright$ Denoised output at current iteration

      $c_1 \leftarrow \eta\sqrt{(1 - \frac{\bar{\alpha}_t}{\bar{\alpha}_s})\frac{1-\bar{\alpha}_s}{1-\bar{\alpha}_t}}$          $\triangleright$ Constants for DDIM
      $c_2 \leftarrow \sqrt{1 - \bar{\alpha}_s - c_1^2}$

      $w_t^2 \Sigma_{\hat{r}_M^t} \leftarrow \widetilde{\mathrm{Cov}}(\hat{r}_M^t)$
      $g \leftarrow ((l_m - \mathcal{A}\cdot\mathcal{D}(\hat{r}_M^t))^\top (w_t^2 \mathcal{A}\Sigma_{\hat{r}_M^t}\mathcal{A}^\top + \sigma_l^2 \mathrm{I})^{-1}\mathcal{A}\frac{\partial\mathcal{D}(\hat{r}_M^t)}{\partial r_M^t})^\top$    $\triangleright$ Likelihood score

      Sample $\epsilon \sim \mathcal{N}(0, \mathrm{I})$
      $r_M \leftarrow \sqrt{\bar{\alpha}_s}\hat{r}_M^t + c_1\epsilon + c_2\epsilon_t + \sqrt{\bar{\alpha}_t}g$          $\triangleright$ Posterior update
   **end for**
   **return** $r_M$          $\triangleright$ Return Estimated Pose sequence

---

**A pertinent question** one may ask is as follows. Given that the user's body shape parameters are available during inference, why not scale the default-body dataset with these body shape parameters? Said differently, applying Eq. 1, the scale-free joint rotations $r_M$ can be scaled—using the available body shape parameters—to regenerate locations $l_M$. This new dataset can then be used to train a CFG model, obviating the need for inverse solvers like `InPose`. Unfortunately, this is possible only if $l_1$ was known (or equal to 0). Otherwise, the mapping between $r_M$ to $l_1$ is non-trivial. As an illustration, consider that the user jumps. Without modeling the dynamics of the human body, it is hard to determine the user's displacement while airborne. This is why pure CFG models are unable to generalize across body shapes, while `InPose`'s inverse guidance formulation does not require new datasets from new users.

## 4 EXPERIMENTS

**Datasets:** All our experiments were performed on AMASS Mahmood et al. (2019), an aggregate of multiple human-pose datasets and the de facto standard for pose estimation/generation today. The data is in the SMPL body model format. Each dataset within AMASS consists of multiple samples, each a sequence of poses at 60, 100, or 120Hz; we resample all data to 60Hz. A fixed default body shape typical of the average male is used for model training. Our experiments follow two dataset protocols, as per our BoDiffusion baseline Castillo et al. (2023):

1. The Transitions Mahmood et al. (2019) and the HumanEVA Sigal et al. (2009) datasets within AMASS were used for testing, while others were used for training.

2. An approximately 90%/10% split for training and testing respectively, on the CMU Carnegie Mellon University, BMLrub Troje (2002), and the HDM05 Müller et al. (2007) datasets.

**Baselines:** We choose the two following SOTA algorithms as baselines. These models accept the 3-point joint locations $l_m(i)$, rotations $r_m(i)$, and the corresponding velocity and angular velocity as inputs; they output the full-body pose of the user for which each was trained. Both the velocity and the angular velocity are computed as linear functions of $l_m$ and $r_m$. The rotation and angular velocities are represented in the 6D representation.

- **AvatarJLM** Zheng et al. (2023) is a conventional neural network-based approach.
- **BoDiffusion** Castillo et al. (2023) is the CFG-based diffusion model that we adopt for `InPose`. In the original BoDiffusion paper, it outputs the local joint angles $\Theta_M(i)$. The authors provide a pretrained model, denoted BoDiffusion(Local), in all our results.
- **BoDiffusion(Global):** We modified BoDiffusion(Local) to output global joint angles $r_M$. This is also evaluated using both $l_m$ and $r_m$ for CFG and is termed BoDiffusion(Global).

**Implementation Details:** We fine-tune the neural network used in BoDiffusion and perform inference using $N = 50$ steps. The original BoDiffusion algorithm implements a DiT Peebles & Xie (2022) based denoiser in a CFG score model framework. Additional details are provided in Appendix B [3].

**Evaluation Metrics:** We use 4 standard metrics from the literature to evaluate the models:

- **Mean Per Joint Position (location) Error(MPJPE)** measures the mean joint location error, in cm, across all joints and poses in the sequence.
- **Mean Per Joint Rotation Error(MPJRE)** measures the mean joint rotation error, in degrees, across all joints and poses. MPJPE captures the scale-dependent error, while MPJRE captures the scale-free error.
- **UPE, LPE:** We also report the joint position error, in cm, for the upper and lower body separately, respectively. These two tell us how well the model can infer upper body versus leg movement, given that all measurement sensors are placed on the upper body.

### 4.1 RESULTS

**Zero-shot generalization across body shape scaling:** Fig. 3(a,b) presents results when the models are trained on a default body shape, and then tested for various body shape scaling factors (including the default). body shapes are varied by changing the scaling factor on the $X$ axis (a value greater than 1.0 on the $X$ axis indicates a proportionally taller human, and vice versa). Note, all bones of the taller person (or shorter) human have been scaled up (or down) by the same factor (we also report separate results where different parts of the body are scaled differently). The root joint translation is also proportional to this scaling factor. The results are performed using Protocol 1. We report location and rotation errors (MPJPE and MPJRE). *Importantly, we divide the estimated MPJPE by the scaling factor*; The MPJRE is obviously scale-free.

The baselines outperform `InPose` in the default case when scale equals 1.0. However, both the scaled MPJPE and the MPJRE (Fig. 3a and 3b) remain almost flat for `InPose` regardless of body shape. This demonstrates the zero-shot nature of our inverse solver, in contrast to the significant degradation observed in the baselines.

**Robustness to measurement noise:** `InPose` is designed to be implicitly robust to location measurement noise as well. We inject zero-mean i.i.d. Gaussian noise into the input location streams and compute the estimation errors, while maintaining the *default* body shape and the rotation measurements. This is an important test for practical applications since real-world wearable sensors—like watches and phones—have difficulty with measurement errors. Fig. 3c shows the location error under increasing Gaussian noise variance (the rotation error is reported in the Appendix). Evidently, `InPose` stays flat while other baselines degrade with noise. This is expected because while BoDiffusion's model is sensitive to location noise, `InPose` uses the location only for inverse guidance, allowing the prior to play an important role in the final pose estimates. In our experiments, we also found that the velocity error in noisy conditions is lower in the case of `InPose` than with

---

[3] We provide the code on our website `https://iclrinpose-crypto.github.io/ICLRInPose/` for reproducability.

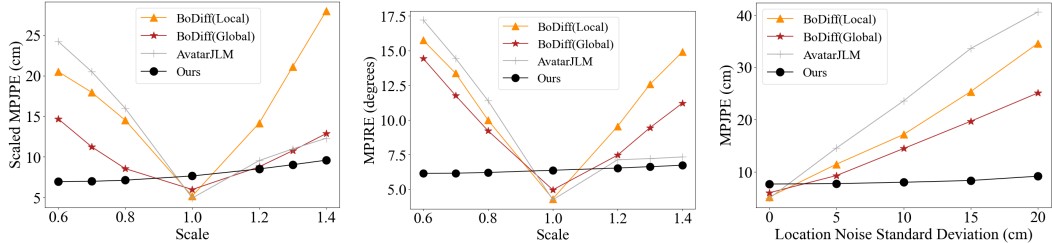

Figure 3: (a) Position error vs. body shape scaling. (b) Rotation error vs. body scale. (c) Position error vs. location noise. All these tests were performed using Protocol 1.

BoDiffusion. The output pose sequences from the baselines have high jitter, indicating the estimated poses are out of distribution.

**Qualitative results with scaling:** Fig.4 presents qualitative comparisons between `InPose` and BoDiffusion(Global), for the default body shape and two scaling factors of $0.6$ and $1.4$. BoDiffusion performs better for the default size, especially in the lower body, but degrades at the task of generalization. The errors are especially prominent in the $0.6$ case, where BoDiffusion predicts the lower body to be in a squatted pose because the measurements are generated by a user of short stature. Since the priors were learned by BoDiffusion from data generated by a user with the default body shape, there is no way to inform the model of this difference. For the $1.4$ case too, BoDiffusion incurs higher torso error.

**Varying relative sizes of body parts:** Table 1 reports results when the body parts are not scaled up or down uniformly; instead, limbs and torso are scaled with different scaling factors.

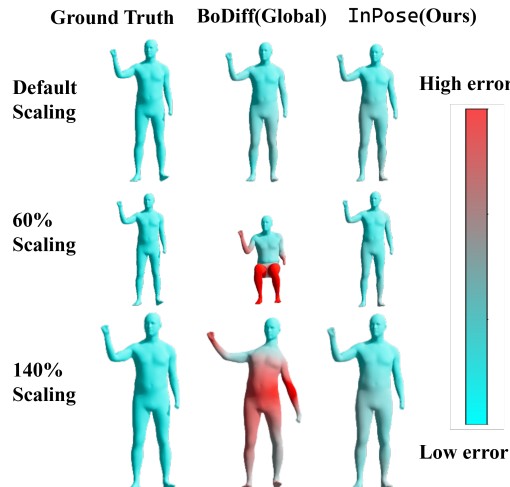

Figure 4: Qualitative results with scaling body shape. The same pose is used for all scales.

This accounts even for outliers in human variations, e.g., basketball players with longer arms, or athletes with longer legs. To create the ground truth data, we scale bone lengths first, which are then used to recompute the joint locations $l_M$ from the scale-free joint rotations $r_M$. Unfortunately, the root translation $l_1$ is a non-trivial function of $r_M$ and the body shape. But in general, we observe that $l_1$ is similar across body shapes that share the same lower body bone lengths. Thus, we preserve both $l_1$ and the lower body shape, and only vary the upper body shape for these tests.

Evident from Table 1 (and more results in Table 3 in the Appendix), the results are aligned with previous graphs. With the default body shape, the baselines outperform `InPose` on all $4$ metrics. This is because the respective neural networks are able to learn a complex non-linear mapping from both the joint rotation $r_m$ and the location inputs $l_m$ to the user's pose since the training body shape (default shape) and the inference body shape are identical. In contrast, `InPose` uses linear constraints using $l_m$ to steer the output towards an estimate of the pose sequence that best explains the input.

However, when bone lengths change, the baselines are misguided by the input locations and therefore do not generalize. `InPose` outperforms both baselines across the MPJRE, MPJPE, and UPE metrics, as inverse location guidance accounts for changes in body shape. Unfortunately, the LPE error remains higher for `InPose` in some cases.

**Using Head translation for CFG:** We also test using the head joint position as an input to the CFG score model. This is termed `InPose` (head) in Table 2. The shape variations compared are less extreme than in Table 1 to be more favorable to the Baselines.

In this comparison, we find that although the MPJRE of `InPose` without head input stays consistent across body shapes, it falls behind the other baselines when tested with relatively minor variation in

Table 1: Algorithm comparison for varying upper body shape. The metrics used are Mean Joint Position Error(MPJPE) in cm, Mean Joint Rotation Error(MPJRE) in degrees, Upper Joint Position Error(UPE) in cm, and Lower Joint Position Error(LPE) in cm. The lower body shape was kept the same, while the upper body bone lengths were scaled.

(a) Results with Upper body shape variation (Protocol 1) (↓ is better)

| Algorithm | Default shape | | | | Upper body ×1.4 | | | | Arms ×1.4, Torso ×0.7 | | | |
|---|---|---|---|---|---|---|---|---|---|---|---|---|
| | MPJPE | MPJRE | UPE | LPE | MPJPE | MPJRE | UPE | LPE | MPJPE | MPJRE | UPE | LPE |
| AvatarJLM | **4.92** | **4.25** | **2.13** | 9.94 | 26.09 | 7.02 | 25.46 | 27.47 | 18.89 | 9.33 | 14.76 | 25.95 |
| BoDiffusion(Local) | 5.16 | 4.32 | 2.36 | **9.72** | 25.69 | 15.35 | 22.79 | 30.21 | 9.98 | 9.24 | 8.05 | 13.33 |
| BoDiffusion(Global) | 5.97 | 4.97 | 2.35 | 11.96 | 13.40 | 11.48 | 10.91 | 17.93 | 7.61 | 7.24 | 5.15 | **11.98** |
| InPose | 7.64 | 6.38 | 3.36 | 14.74 | **9.15** | **6.71** | **4.80** | **16.31** | **7.45** | **6.52** | **3.23** | 14.6 |

(b) Results with Upper body shape variation (Protocol 2) (↓ is better)

| Algorithm | Default shape | | | | Upper body ×1.4 | | | | Arms ×1.4, Torso ×0.7 | | | |
|---|---|---|---|---|---|---|---|---|---|---|---|---|
| | MPJPE | MPJRE | UPE | LPE | MPJPE | MPJRE | UPE | LPE | MPJPE | MPJRE | UPE | LPE |
| AvatarJLM | **3.54** | 3.11 | **1.49** | **6.92** | 27.13 | 8.36 | 26.02 | 29.21 | 20.32 | 9.34 | 15.83 | 27.98 |
| BoDiffusion(Local) | 3.59 | **2.68** | 1.51 | 7.0 | 26.12 | 14.51 | 21.34 | 33.62 | 10.13 | 8.41 | 8.13 | **13.63** |
| BoDiffusion(Global) | 4.90 | 3.45 | 1.90 | 9.79 | 17.39 | 12.71 | 13.59 | 23.77 | 9.99 | 8.78 | 7.25 | 14.85 |
| InPose | 7.53 | 4.73 | 2.9 | 15.04 | **8.94** | **4.73** | **3.97** | **16.85** | **6.96** | **4.77** | **2.60** | 14.17 |

bone lengths compared to the default body shape. However, by simply adding the head position as a CFG input to the diffusion model, `InPose` (head) outperforms all baselines across the tested body shape variations in nearly all metrics. Furthermore, we see considerable improvement in lower body motion estimation as evidenced by the LPE metrics.

Table 2: Algorithm comparison for varying upper body shape using Protocol 1. `InPose` (head) augments `InPose` by using the head translation input for CFG. The Shape variation in this table is less extreme than in Table 1.

| Algorithm | Upper body ×0.85 | | | | Upper body ×1.17 | | | | Arms ×0.85, torso ×1.17 | | | |
|---|---|---|---|---|---|---|---|---|---|---|---|---|
| | MPJPE | MPJRE | UPE | LPE | MPJPE | MPJRE | UPE | LPE | MPJPE | MPJRE | UPE | LPE |
| BoDiffusion(Local) | 7.44 | 7.47 | 5.03 | **11.57** | 11.72 | 8.76 | 9.21 | 15.82 | 6.27 | 4.91 | 3.64 | **10.58** |
| BoDiffusion(Global) | 6.76 | 7.79 | 3.59 | 12.16 | 8.18 | 6.99 | 5.57 | 12.66 | 6.8 | 5.7 | 3.36 | 12.49 |
| InPose | 7.13 | 6.26 | 2.87 | 14.25 | 8.25 | 6.51 | 3.95 | 15.37 | 7.8 | 6.33 | 3.46 | 14.95 |
| InPose (head) | **6.08** | **5.06** | **2.4** | 12.06 | **5.92** | **4.51** | **2.63** | **11.29** | **5.71** | **4.5** | **2.37** | 11.16 |

**Ablation: 6D versus rotation matrices:** Recall that `InPose` needed to tackle the non-linearity from the $\mathcal{D}(.)$ function, which was needed to convert 6D representation to matrices. A natural question is: was it necessary to use 6D at all? Figure 5 shows the importance of 6D over $3 \times 3$ rotation matrices. We train two unconditional UNET-based diffusion models—one using the 6D representation and the other using 3D rotation matrices. These models generate 64-frame human pose samples as global joint angles in their respective representations. These poses are then used to generate sequences of body meshes. The raw rotation matrix meshes (labeled RoT-Raw) from the Rotation

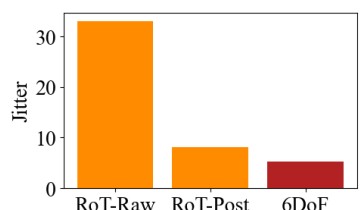

Figure 5: 6D vs. rotation matrix representation for Diffusion compared using Jitter (↓ is better).

matrix UNET exhibit high jitter. In contrast, the 6D meshes(labeled 6D) from the 6D UNET have much lower jitter. This is primarily because the raw output rotation matrices are not unitary. In fact, postprocessing the rotation matrices using $\mathcal{D}(.)$ on the first 2 columns (labeled RoT-Post) considerably lowers the jitter.

More qualitative results are shown in the Appendix, and we provide some animations on our website: `https://iclrinpose-crypto.github.io/ICLRInPose/`.

## 5 RELATED WORK

**Deep-learning based methods for pose-tracking:** Deep learning has achieved significant success in estimating pose from a sparse set of measurements. Aliakbarian et al. (2023); Du et al. (2023);

Yuan et al. (2023) all use HMD-based location and rotation sensors to estimate pose and translation. Nearly all these works focus on using sensor information from the head and the two wrists. Jiang et al. (2022) estimate pose by first using a Transformer encoder to estimate local joint angles, and then estimates translation by fitting the generated head translation to the head location sensor input. Castillo et al. (2023) use a CFG diffusion-based approach to estimate pose. Most of the above-mentioned approaches are specifically trained for a single user's body parameters, which comes at the cost of worse generalizability. One work that does generalize across users, Aliakbarian et al. (2022) jointly trains sensor inputs and bone length parameters in a flow-based generative model framework. But this algorithm requires jointly training pose and a large number of body shapes in order to generalize. In contrast, our work can directly accept any set of body bone parameters without requiring any bone shape generalization training.

A large number of works Huang et al. (2018); Mollyn et al. (2023) focus on pose and translation prediction using a sparse set of IMUs that provide acceleration and orientation data from the joints to which they are attached. Yi et al. (2021) use a cascaded sequence of Neural networks to predict pose from 6 IMU sensors. Yi et al. (2022) incorporate physics-based dynamics constraints on the user's joint motion to improve pose estimation accuracy.

**Human Motion synthesis:** A closely related topic to pose estimation is pose synthesis Raab et al. (2023); Tevet et al. (2025). This usually involves training a generative model on a human motion dataset, along with textual labels, to generate motion either based on a prompt or unconditionally. Shafir et al. (2024) use a trained motion synthesis model as a prior for more complex tasks, such as motion blending and multi-person interactive motion.

Some of these textual models also accept other inputs to serve as guidance for the generated motion Tessler et al. (2024); Diller & Dai (2024). Xie et al. (2024) generate human motion from textual prompts using a diffusion model, but can also use a gradient-descent-based inverse guidance method to specify motion trajectories of various joints. Their work is closely related to ours, but they also require a Transformer-based realism-guidance module that encodes joint location-control signals.

## 6 LIMITATIONS AND FUTURE WORK

The biggest limitation of `InPose` is that the root translation isn't directly incorporated into the algorithm, which reduces its capability to infer lower body pose. Thus, the algorithm can only use the prior to infer translation and, thereby, infer lower body movement. This limitation can be addressed by using the head position input in the CFG score model as in `InPose` (Head). This is, unfortunately, at the cost of reducing the inherent flexibility of our inverse guidance approach.

Another major limitation is that `InPose` cannot outperform the baselines in the default body shape scenario. A reason for this may be that the Gaussian approximation in Theorem 1 may not be sound.

Furthermore, `InPose` requires the user's body bone lengths during inference. An interesting direction to pursue is blindly determining the user's bone lengths without an explicit calibration procedure.

## 7 CONCLUSIONS

We propose `InPose`, a diffusion-based model that estimates the user's 3D fully-body pose sequence from 3 sensor measurements. By decomposing poses into a scale-free and a scale-dependent component, we find a pathway to an inverse problem formulation that in turn enables zero-shot generalization. As a result, any new body shape or shape need not be re-trained with personalized data; `InPose` is able to guide the diffusion-based prior by computing whether the samples from the prior are consistent with the joint location sensor measurements. There is room for improvement at least on two fronts, namely in outperforming the baselines for the default sizes for which they are trained, and in improving lower body errors by better modeling the physics of leg movements. We believe these are rich problems for future research.

## 8 ACKNOWLEDGEMENTS

We thank the anonymous reviewers for their invaluable feedback. This work was partially supported by NSF #2008338, #1909568, #2148583, and #MRI-2018966. This work used DELTA at NCSA through allocation CIS230230 from the Advanced Cyberinfrastructure Coordination Ecosystem: Services & Support (ACCESS) program, which is supported by U.S. National Science Foundation grants #2138259, #2138286, #2138307, #2137603, and #2138296.

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

## A  PROOF OF THEOREM 1

**Theorem 1.** *We are given a well-trained score model $\epsilon_\theta$, that learns the score function $\epsilon_t \leftarrow \epsilon_\theta(r_M^t, t, r_m)$, and denoises $\hat{r}_M^t \leftarrow \frac{r_M^t - \sqrt{1-\bar{\alpha}_t}\epsilon_t}{\sqrt{\bar{\alpha}_t}}$. If the model ensures that $||\hat{r}_j^{t,1:3}|| = ||\hat{r}_j^{t,4:6}|| = 1$, $\langle \hat{r}_j^{t,1:3}, \hat{r}_j^{t,4:6} \rangle = 0$, $\forall j \in M$ then $p_t(\mathcal{D}(r_M^0)|r_M^t) \approx \mathcal{N}(\mathcal{D}(\hat{r}_M^t), w_t^2 \Sigma_{\hat{r}_M^t})$ where $\Sigma_{\hat{r}_M^t}$ is a positive definite matrix.*

*Proof.* (Sketch)We will prove the validity of the Gaussian approximation for a single joint rotation $r_j^t$. This result can then be naturally extended to $r_M^t$ because each joint rotation is independent in the global reference frame. From ΠGDM, we approximate $p_t(r_M^0|r_M^t) \approx \mathcal{N}(\hat{r}_M^t, w_t^2 I)$. This implies that every element $r_j^{0,k}$, $\forall k \in \{1:6\}$, $\forall j \in M \sim \mathcal{N}(\hat{r}_j^{t,k}, w_t)$ are i.i.d. Gaussian Random variables.

The mapping $\bar{\mathcal{D}}(r_j^0) = R_j^0$ is defined as:

$$
\begin{aligned}
c^1 &= \left[ r_j^{1:3} \right]^\top, & \bar{c}^1 &= \frac{c^1}{||c^1||} \\
c^2 &= \left[ r_j^{4:6} \right]^\top - \text{proj}_{\bar{c}^1}\left( \left[ r_j^{4:6} \right]^\top \right), & \bar{c}^2 &= \frac{c^2}{||c^2||} \\
\bar{c}^3 &= \bar{c}^1 \times \bar{c}^2 \\
R_j &= \begin{bmatrix} \bar{c}^1 & \bar{c}^2 & \bar{c}^3 \end{bmatrix}
\end{aligned}
\tag{11}
$$

By the definition of $\bar{\mathcal{D}}(r_j^0) = R_j^0$, the unit norm constraints $||\hat{r}_j^{t,1:3}|| = ||\hat{r}_j^{t,3:6}|| = 1$, and $\langle \hat{r}_j^{t,1:3}, \hat{r}_j^{t,3:6} \rangle = 0$, the elements of the first two columns, $R_j^{0,(1:2,1:3)} \sim \mathcal{N}(\hat{R}_j^{t,(l,m)}, w_t^2)$ are also Gaussian random variables that are uncorrelated. Now, the elements of the third column of each $R_j^{0,(3,1:3)}$ are the result of the cross-product $r_j^{0,1:3} \times r_j^{0,3:6}$. Let's first discuss $R_j^{0,(3,2)} = (r_j^{0,3} r_j^{0,4} - r_j^{0,1} r_j^{0,6})$. The mean of this random variable is:

$$
\begin{aligned}
\mathbb{E}[r_j^{0,3} r_j^{0,4} - r_j^{0,1} r_j^{0,6}] &= \mathbb{E}[r_j^{0,3} r_j^{0,4}] - \mathbb{E}[r_j^{0,1} r_j^{0,6}] & (12) \\
&= \mathbb{E}[r_j^{0,3}]\mathbb{E}[r_j^{0,4}] - \mathbb{E}[r_j^{0,1}]\mathbb{E}[r_j^{0,6}] & (13) \\
&= \hat{r}_j^{t,3}\hat{r}_j^{t,4} - \hat{r}_j^{t,1}\hat{r}_j^{t,6} & (14)
\end{aligned}
$$

Thus, all the elements of the third column $R_j^{0,(3,1:3)}$ have their mean given by the respective cross-product terms. Next, we can compute the variance as:

$$
\text{Var}[r_j^{0,3} r_j^{0,4} - r_j^{0,1} r_j^{0,6}] = \mathbb{E}[(r_j^{0,3} r_j^{0,4} - r_j^{0,1} r_j^{0,6})^2] - \mathbb{E}[r_j^{0,3} r_j^{0,4} - r_j^{0,1} r_j^{0,6}]^2 \tag{15}
$$

Treating the terms separately, we get

$$
\begin{aligned}
&\mathbb{E}[(r_j^{0,3} r_j^{0,4} - r_j^{0,1} r_j^{0,6})^2] \\
&= \mathbb{E}[(r_j^{0,3} r_j^{0,4})^2] + \mathbb{E}[(r_j^{0,1} r_j^{0,6})^2] - 2\mathbb{E}[(r_j^{0,3} r_j^{0,4} r_j^{0,1} r_j^{0,6})] & (16) \\
&= \mathbb{E}[(r_j^{0,3})^2]\mathbb{E}[(r_j^{0,4})^2] + \mathbb{E}[(r_j^{0,1})^2]\mathbb{E}[(r_j^{0,6})^2] - 2\mathbb{E}[(r_j^{0,3} r_j^{0,4} r_j^{0,1} r_j^{0,6})] & (17) \\
&= (w_t^2 + (\hat{r}_j^{t,3})^2)(w_t^2 + (\hat{r}_j^{t,4})^2) + (w_t^2 + (\hat{r}_j^{t,1})^2)(w_t^2 + (\hat{r}_j^{t,6})^2) - 2\hat{r}_j^{t,3}\hat{r}_j^{t,4}\hat{r}_j^{t,1}\hat{r}_j^{t,6} & (18) \\
&= 2w_t^4 + w_t^2((\hat{r}_j^{t,3})^2 + (\hat{r}_j^{t,4})^2) + (\hat{r}_j^{t,1})^2) + (\hat{r}_j^{t,6})^2)) + (\hat{r}_j^{t,3}\hat{r}_j^{t,4})^2 + (\hat{r}_j^{t,1}\hat{r}_j^{t,6})^2 - 2\hat{r}_j^{t,3}\hat{r}_j^{t,4}\hat{r}_j^{t,1}\hat{r}_j^{t,6} & (19)
\end{aligned}
$$

using the independence property and the definition of variance. Next,

$$\mathbb{E}[r_j^{0,3} r_j^{0,4} - r_j^{0,1} r_j^{0,6}]^2 = \mathbb{E}[r_j^{0,3} r_j^{0,4}]^2 + \mathbb{E}[r_j^{0,1} r_j^{0,6}]^2 - 2\mathbb{E}[(r_j^{0,3} r_j^{0,4} r_j^{0,1} r_j^{0,6})] \tag{20}$$

$$= (\hat{r}_j^{t,3} \hat{r}_j^{t,4})^2 + (\hat{r}_j^{t,1} \hat{r}_j^{t,6})^2 - 2\hat{r}_j^{t,3} \hat{r}_j^{t,4} \hat{r}_j^{t,1} \hat{r}_j^{t,6} \tag{21}$$

Substituting the above terms into Eqn. 15, we get

$$\mathrm{Var}[r_j^{0,3} r_j^{0,4} - r_j^{0,1} r_j^{0,6}] = w_t^2 (2w_t^2 + ((\hat{r}_j^{t,3})^2 + (\hat{r}_j^{t,4})^2) + (\hat{r}_j^{t,1})^2) + (\hat{r}_j^{t,6})^2)) \tag{22}$$

Finally, we compute each of the covariances of the third column elements $R_j^{0,(3,1:3)}$. To compute $\mathrm{Cov}[R_j^{0,(3,2)}, r_j^{0,1}]$:

$$\mathrm{Cov}[r_j^{0,3} r_j^{0,4} - r_j^{0,1} r_j^{0,6}, r_j^{0,1}]$$

$$= \mathrm{Cov}[r_j^{0,3} r_j^{0,4}, r_j^{0,1}] - \mathrm{Cov}[(r_j^{0,1})^2 r_j^{0,6}] \tag{23}$$

$$= \mathbb{E}[r_j^{0,3} r_j^{0,4} r_j^{0,1}] - \mathbb{E}[r_j^{0,3} r_j^{0,4}]\mathbb{E}[r_j^{0,1}] - \mathbb{E}[(r_j^{0,1})^2 r_j^{0,6}] + \mathbb{E}[r_j^{0,1} r_j^{0,6}]\mathbb{E}[r_j^{0,1}] \tag{24}$$

$$= 0 - (w_t^2 + (\hat{r}_j^{t,1})^2)\hat{r}_j^{t,6} + (\hat{r}_j^{t,1})^2 \hat{r}_j^{t,6} \tag{25}$$

$$= -w_t^2 \hat{r}_j^{t,6} \tag{26}$$

and $\mathrm{Cov}[R_j^{0,(3,1)}, R_j^{0,(3,2)}]$:

$$\mathrm{Cov}[r_j^{0,2} r_j^{0,6} - r_j^{0,3} r_j^{0,5}, r_j^{0,3} r_j^{0,4} - r_j^{0,1} r_j^{0,6}]$$

$$= 0 - \mathrm{Cov}[r_j^{0,3} r_j^{0,5}, r_j^{0,3} r_j^{0,4}] + 0 - \mathrm{Cov}[r_j^{0,2} r_j^{0,6}, r_j^{0,1} r_j^{0,6}] \tag{27}$$

$$= \mathbb{E}[r_j^{0,3} r_j^{0,5}]\mathbb{E}[r_j^{0,3} r_j^{0,4}] - \mathbb{E}[(r_j^{0,3})^2 r_j^{0,5} r_j^{0,4}] + \mathbb{E}[r_j^{0,2} r_j^{0,6}]\mathbb{E}[r_j^{0,1} r_j^{0,6}] - \mathbb{E}[(r_j^{0,6})^2 r_j^{0,1} r_j^{0,2}] \tag{28}$$

$$= 0 - w_t^2 \hat{r}_j^{t,4} \hat{r}_j^{t,5} + 0 - w_t^2 \hat{r}_j^{t,1} \hat{r}_j^{t,2} \tag{29}$$

$$= -w_t^2 (\hat{r}_j^{t,4} \hat{r}_j^{t,5} + \hat{r}_j^{t,1} \hat{r}_j^{t,2}) \tag{30}$$

The list of variances is the following:

$$\mathrm{Var}[R_j^{0,(3,1)}] = w_t^2 (2w_t^2 + (\hat{r}_j^{t,2})^2 + (\hat{r}_j^{t,6})^2 + (\hat{r}_j^{t,5})^2 + (\hat{r}_j^{t,3})^2)$$

$$\mathrm{Var}[R_j^{0,(3,2)}] = w_t^2 (2w_t^2 + (\hat{r}_j^{t,3})^2 + (\hat{r}_j^{t,4})^2 + (\hat{r}_j^{t,1})^2 + (\hat{r}_j^{t,6})^2)$$

$$\mathrm{Var}[R_j^{0,(3,3)}] = w_t^2 (2w_t^2 + (\hat{r}_j^{t,1})^2 + (\hat{r}_j^{t,5})^2 + (\hat{r}_j^{t,4})^2 + (\hat{r}_j^{t,2})^2)$$

and the covariances are:

$\mathrm{Cov}[R_j^{0,(3,1)}, r_j^{0,2}] = w_t^2 \hat{r}_j^{t,6} \qquad \mathrm{Cov}[R_j^{0,(3,2)}, r_j^{0,1}] = -w_t^2 \hat{r}_j^{t,6} \qquad \mathrm{Cov}[R_j^{0,(3,3)}, r_j^{0,1}] = w_t^2 \hat{r}_j^{t,5}$

$\mathrm{Cov}[R_j^{0,(3,1)}, r_j^{0,3}] = -w_t^2 \hat{r}_j^{t,5} \qquad \mathrm{Cov}[R_j^{0,(3,2)}, r_j^{0,3}] = w_t^2 \hat{r}_j^{t,4} \qquad \mathrm{Cov}[R_j^{0,(3,3)}, r_j^{0,2}] = -w_t^2 \hat{r}_j^{t,4}$

$\mathrm{Cov}[R_j^{0,(3,1)}, r_j^{0,5}] = -w_t^2 \hat{r}_j^{t,3} \qquad \mathrm{Cov}[R_j^{0,(3,2)}, r_j^{0,4}] = w_t^2 \hat{r}_j^{t,3} \qquad \mathrm{Cov}[R_j^{0,(3,3)}, r_j^{0,4}] = -w_t^2 \hat{r}_j^{t,2}$

$\mathrm{Cov}[R_j^{0,(3,1)}, r_j^{0,6}] = w_t^2 \hat{r}_j^{t,2} \qquad \mathrm{Cov}[R_j^{0,(3,2)}, r_j^{0,6}] = -w_t^2 \hat{r}_j^{t,1} \qquad \mathrm{Cov}[R_j^{0,(3,3)}, r_j^{0,5}] = w_t^2 \hat{r}_j^{t,1}$

$$\mathrm{Cov}[R_j^{0,(3,1)}, R_j^{0,(3,2)}] = -w_t^2 (\hat{r}_j^{t,1} \hat{r}_j^{t,2} + \hat{r}_j^{t,4} \hat{r}_j^{t,5})$$

$$\mathrm{Cov}[R_j^{0,(3,2)}, R_j^{0,(3,3)}] = -w_t^2 (\hat{r}_j^{t,2} \hat{r}_j^{t,3} + \hat{r}_j^{t,5} \hat{r}_j^{t,6})$$

$$\mathrm{Cov}[R_j^{0,(3,3)}, R_j^{0,(3,1)}] = -w_t^2 (\hat{r}_j^{t,1} \hat{r}_j^{t,3} + \hat{r}_j^{t,4} \hat{r}_j^{t,6})$$

while the terms that have been omitted are all $0$.

Now that we have the respective variances and covariances, we can build the positive definite covariance matrix for $p_t(\bar{\mathcal{D}}(r_j^0) = \text{vec}(R_j^0)|r_j^t)$, which at diffusion step $t$ is $w_t^2 \Sigma_{\hat{r}_j^t}$. To show that $\Sigma_{\hat{r}_j^t}$ is a positive definite matrix, we use Sylvesters criterion Gilbert (1991). It states that a symmetric matrix $\Sigma \in \mathbb{R}^{N \times N}$ is positive definite if each upper left corner matrix of sizes $n \in \{1, \dots, N\}$ has a positive determinant. Since $\Sigma_{\hat{r}_j^t}$ is a relatively large matrix of size $9 \times 9$, we use SymPy Meurer et al. (2017) to compute each corner matrix determinant.

We find that each determinant equals a positive number that depends on $w_t^2$. The first 6 corner matrices trivially have determinant 1 since they are all identity matrices. The determinant for $n = \{7, 8, 9\}$ (termed $\Sigma_{\hat{r}_j^t}^{n \times n}$) are the following:

$$\det(\Sigma_{\hat{r}_j^t}^{7 \times 7}) = 2w_t^2 \qquad \det(\Sigma_{\hat{r}_j^t}^{8 \times 8}) = (2w_t^2)^2 \qquad \det(\Sigma_{\hat{r}_j^t}) = (2w_t^2)^3$$

thus proving that $\Sigma_{\hat{r}_j^t}$ is positive definite.

Since we assume every rotation in $r_M^t$ is independent of each other, when we combine the $w_t^2 \Sigma_{\hat{r}_j^t}$ for each rotation, we get a large positive definite matrix $w_t^2 \Sigma_{\hat{r}_M^t}$. Thus, we can approximate the distribution $p_t(\mathcal{D}(r_M^0)|r_M^t)$ as a Gaussian using the mean and the derived positive definite matrix.

$\square$

## B  Implementation Details

As stated earlier, we use BoDiffusion as the base Diffusion model for `InPose`. The conditional inputs for the CFG score model are the joint rotations, locations, the joint velocities, and the angular velocities from the 3 measured joints. The rotations are provided using the 6D representation. The output of the network is the 22 joint rotations $\Theta_M$ in the local reference frame, expressed in 6D. The model is designed to output a frame of 41 samples. For sequences larger than 41 samples, it uses an overlap of 20 samples between successive frames.

Since we require the model to output joint angles $r_M$ in the global frame, we fine-tune it on our training datasets to produce global rotation angles. We use the weights provided by the BoDiffusion authors to initialise fine-tuning. We also tune the CFG model to use only joint rotations and angular velocities by training it on a subset of samples with zeroed-out joint locations and velocities. During inverse-guidance-based inference, we can similarly zero out the location and velocity CFG inputs to the diffusion network.

This model has about 22M parameters. Finetuning is done using the DDPM framework for $N = 1000$ steps. Inference is done for $N = 50$ steps using DDIM, for both pure CFG-based guidance as well as `InPose`'s inverse guidance. We used an Nvidia RTX Titan GPU for 2 days of training with a batch size of 256.

**Inverse Guidance:** For the $\Pi$GDM-based inverse guidance term $\nabla_{r^t} \log p_t(l_m|r_M^t)$, we used an additional scale parameter, which we found was useful in improving performance for all metrics. Increasing this term led to minor increases in MPJPE performance at the cost of worse MPJRE error. Secondly, since the positive definite matrix $\Sigma_{\hat{r}_M^t}$ is very large, its usage substantially slows down run-time. We found that setting $\Sigma_{\hat{r}_M^t}$ to simply the identity matrix performs reasonably well.

## C  More Results

**Performance with varying upper body(Part2):** We show more results with varying upper body shape in Table 3. We once again see that `InPose` is able to generalize better to changes in relative bone lengths, but lags behind the other baselines in lower body pose estimation.

**Performance with scaling body shape on Protocol 2:** Figures 6a and 6b are the performance results from scaling body shape using Protocol 2. The tests were conducted in a manner similar to what

Table 3: Algorithm comparison for varying upper body shape. The metrics used are Mean Joint Position Error(MPJPE) in cm, Mean Joint Rotation Error(MPJRE) in degrees, Upper Joint Position Error(UPE) in cm, and Lower Joint Position Error(LPE) in cm. The lower body shape was kept the same, while the upper body bone lengths were scaled.

(a) Results with Upper body shape variation (Protocol 1) (↓ is better)

| Algorithm | Upper body ×0.7 | | | | Arms ×1.4 | | | | Arms ×0.7 | | | |
|---|---|---|---|---|---|---|---|---|---|---|---|---|
| | MPJPE | MPJRE | UPE | LPE | MPJPE | MPJRE | UPE | LPE | MPJPE | MPJRE | UPE | LPE |
| AvatarJLM | 20.59 | 10.33 | 19.02 | 23.32 | 9.21 | 7.21 | 7.79 | **11.84** | 7.90 | 8.15 | 5.88 | **11.46** |
| BoDiffusion(Local) | 9.00 | 11.29 | 6.68 | 13.2 | 15.57 | 11.99 | 13.32 | 19.29 | 9.36 | 8.84 | 7.12 | 13.27 |
| BoDiffusion(Global) | 7.44 | 10.56 | 4.35 | **12.81** | 9.51 | 9.02 | 7.18 | 13.66 | 7.83 | 8.96 | 4.89 | 12.88 |
| InPose | **6.67** | **6.17** | **2.42** | 13.80 | **8.25** | **6.67** | **3.97** | 15.4 | **7.22** | **6.20** | **2.92** | 14.35 |

(b) Results with Upper body shape variation (Protocol 2) (↓ is better)

| Algorithm | Upper body ×0.7 | | | | Arms ×1.4 | | | | Arms ×0.7 | | | |
|---|---|---|---|---|---|---|---|---|---|---|---|---|
| | MPJPE | MPJRE | UPE | LPE | MPJPE | MPJRE | UPE | LPE | MPJPE | MPJRE | UPE | LPE |
| AvatarJLM | 21.63 | 9.06 | 19.86 | 24.56 | 10.30 | 8.02 | 8.57 | **13.36** | 7.44 | 6.67 | 6.17 | **9.65** |
| BoDiffusion(Local) | 7.72 | 9.14 | 6.06 | **10.67** | 15.93 | 11.11 | 12.77 | 21.11 | 7.61 | 6.95 | 6.30 | 9.91 |
| BoDiffusion(Global) | 9.17 | 9.32 | 6.72 | 13.38 | 13.19 | 10.76 | 9.83 | 18.94 | 9.19 | 7.17 | 7.13 | 12.63 |
| InPose | **6.53** | **4.75** | **2.11** | 13.84 | **7.80** | **4.74** | **3.21** | 15.24 | **7.34** | **4.78** | **2.65** | 14.97 |

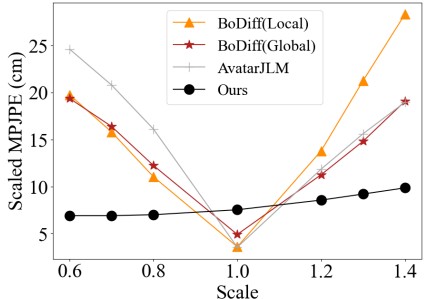
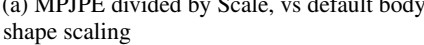

(a) MPJPE divided by Scale, vs default body shape scaling

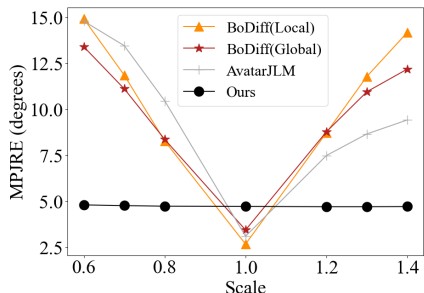

(b) MPJRE vs body scale

Figure 6: Performance with body-size scaling using Protocol 2.

is described in Section 4.1. We once again see that InPose performs worse than the baselines in the default case, where the same body shape used during training is also used for testing. However, when the default body shape is scaled, InPose outperforms the baselines. MPJRE, which measures scale-free performance, remains constant, while the scale-dependent MPJPE scales proportionally with scaling.

**Robustness to measurement noise(cntd):** Here are the rotation error results from the robustness study we performed in the Section 4.1. As stated earlier, we injected zero-mean i.i.d. Gaussian noise into the input location streams and computed the estimation errors, while maintaining the *default* body shape. Fig. 7 shows the rotation error(MPJRE) under increasing Gaussian noise variance in the location measurements. As with the location error, InPose stays flat while other baselines degrade with noise. Since the ΠGDM inverse guidance objective is formulated to be robust to noise, the prior is able to synthesize poses that are realistic while also obeying the guidance provided by the location inputs.

**Errors in joint length estimates:** Another study involves illustrating InPose's sensitivity to errors in the joint length estimates, where we show how the MPJPE and MPJRE worsen when there is an error in our knowledge of the user's bone lengths. We add white Gaussian noise to the true bone lengths while constructing our measurement matrix $\mathcal{A}$. The bone length parameters are set to the default body shape, and the measurements $y_m$ are unaltered.

Figure 8 shows the results for this experiment. We see that InPose is quite sensitive to bone length estimation error. The algorithm can tolerate low errors within 1 cm, but begins to diverge any higher

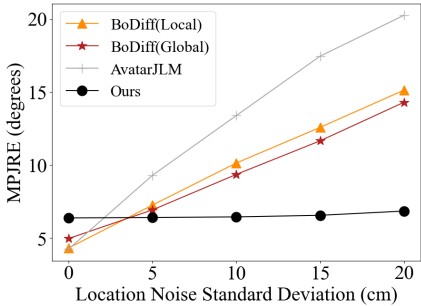

Figure 7: Rotation error vs location noise.

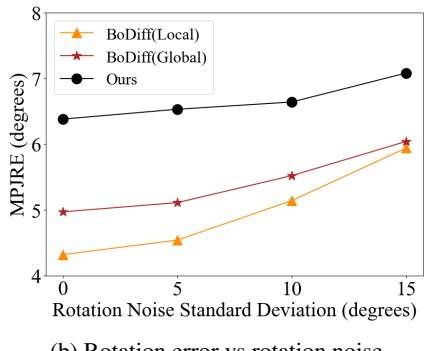

Figure 8: Performance with joint length error. The left axis is the MPJPE, and the right axis is the MPJRE.

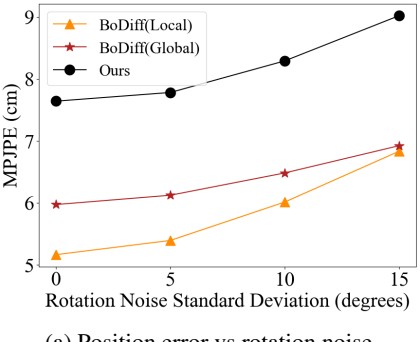

(a) Position error vs rotation noise

(b) Rotation error vs rotation noise

Figure 9: Performance with additive white noise in rotation measurements.

than that. Making the algorithm robust to bone length estimation errors will be looked at for future work.

**Robustness to rotation noise:** We conduct another robustness study with rotation measurements, similar to the location error study described in Section 4.1. Here, we add Gaussian noise to the rotation measurements while keeping the location measurements noise-free. We then compare the `InPose` with the two diffusion-based baselines. The results are summarized in Figure 9. We find that all the diffusion-based algorithms are relatively robust to rotation measurement noise, with both the rotation and position errors rising steadily as rotation noise increases.

**Need for using location measurements:** An important question is whether purely using the scale-free rotations $r_m$ is better than including the scale-dependent location measurements $l_m$. Since $r_m$ is scale-free, we could potentially use only $r_m$ to estimate pose. As an ablation study, we compare BoDiffusion(Global) and `InPose`, against BoDiffusion(Global) without using $l_m$ for CFG. Since BoDiffusion(Global) was trained to accept only $r_m$ or both $\{r_m, l_m\}$ for CFG-based conditioning, this serves as an apples-to-apples comparison.

Table 4 shows the results of this comparison using Protocol 1, using the default body shape to generate $l_m$. BoDiffusion(Global) with $l_m$ for CFG performs the best amongst the three algorithms. `InPose` is next, performing much better than BoDiffusion(Global) without $l_m$, illustrating the importance of $l_m$ for pose estimation.

Table 4: Comparison between `InPose`, BoDiffusion(Global) and BoDiffusion(Global) with no $l_m$ as input for CFG using Protocol 1

| Metric | InPose | BoDiffusion(Global) | BoDiffusion(Global) no $l_m$ |
|---|---|---|---|
| MPJPE(cm) | 7.64 | **5.97** | 15.98 |
| MPJRE(°) | 6.38 | **4.97** | 8.71 |

**Using Inverse-Guidance on Local Joint Angle Representation:** Considering that the BoDiffusion(Local) model outperforms the BoDiffusion(Global) model, an important question is—why not use the local joint angle output $\Theta_M$ for inverse guidance? In earlier comparisons, we saw that Local prediction achieves better lower-body (LPE) performance.

To compare Inverse Guidance performance for Local joint angle prediction with Global joint angle prediction, we fine-tune BoDiffusion(Local) in a manner similar to `InPose` (head) as described in B, such that it accepts head position along with all 3 rotation angles as input for the CFG score, but the hand positions are zeroed out. We use a $\Pi$GDM-based inverse guidance after computing the global joint angles using the local angle output from the score model, which we term Local($\Pi$GDM). We also use an L2-based gradient descent method similar to that in Xie et al. (2024) to perform inverse guidance using joint location. We term this method Local(Gradient Descent) and Global(Gradient Descent) based on whether the score model predicts local or global joint angles, respectively.

Table 5: Comparison with Inverse Guidance for Local Joint Angle Prediction using Protocol 1

| Algorithm | MPJPE(cm) | MPJRE(°) | UPE(cm) | LPE(cm) |
|---|---|---|---|---|
| Local(Gradient Descent) | **5.51** | 5.25 | 3.32 | **9.47** |
| Local($\Pi$GDM) | 5.82 | 5.16 | 2.48 | 11.27 |
| Global(Gradient Descent) | 5.58 | 4.97 | 2.71 | 10.14 |
| `InPose` (head) | 5.85 | **4.69** | **2.47** | 11.34 |

The results are summarized in Table 5. The Local(Gradient Descent) is once again better than `InPose` and its variants at predicting lower body motion, resulting in lower MPJPE. However, this is at the cost of worse upper body performance and higher overall joint angle error. `InPose`, which uses $\Pi$GDM was specifically chosen in our work for zero-shot pose prediction because of better upper body inverse guidance, especially when head position isn't used for the CFG score model.

**Runtime Performance:** We next test the runtime of each algorithm in Table 6, reporting the number of samples estimated per second for each. Each algorithm was tested on an RTX Titan. Given that we operate at 60 frames per second across all baselines, all algorithms can run in real time, although some latency is expected.

Table 6: Runtime performance comparison between `InPose`, BoDiffusion and AvatarJLM

| Algorithm | Runtime (samples /sec) |
|---|---|
| AvatarJLM | $\sim 102$ |
| BoDiffusion | $\sim 392$ |
| `InPose` | $\sim 229$ |

**More Qualitative Results:** We show another qualitative sample in Figures 10 and 11, comparing `InPose` with the baseline algorithms in a running pose. Once again, we keep the same relative body shape and modify only the scale. `InPose` falls behind the baselines when it comes to estimating lower body pose, especially the feet. However, it generalizes across all body scales tested, with error at the arms lower than that of the baseline algorithms.

Unfortunately, `InPose` without head position as CFG input catastrophically fails in some scenarios when predicting lower body motion, as seen in Figure 12. But this is mitigated by using the head position as input to the CFG score model, as in `InPose` (head).

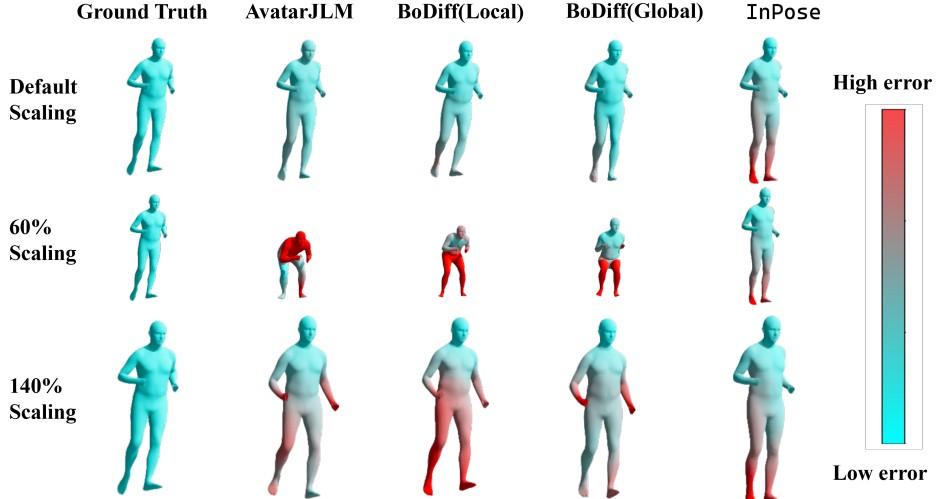

Figure 10: More qualitative results comparing `InPose` with the Baselines with varying body scale. Relative body shape and pose have been kept constant.

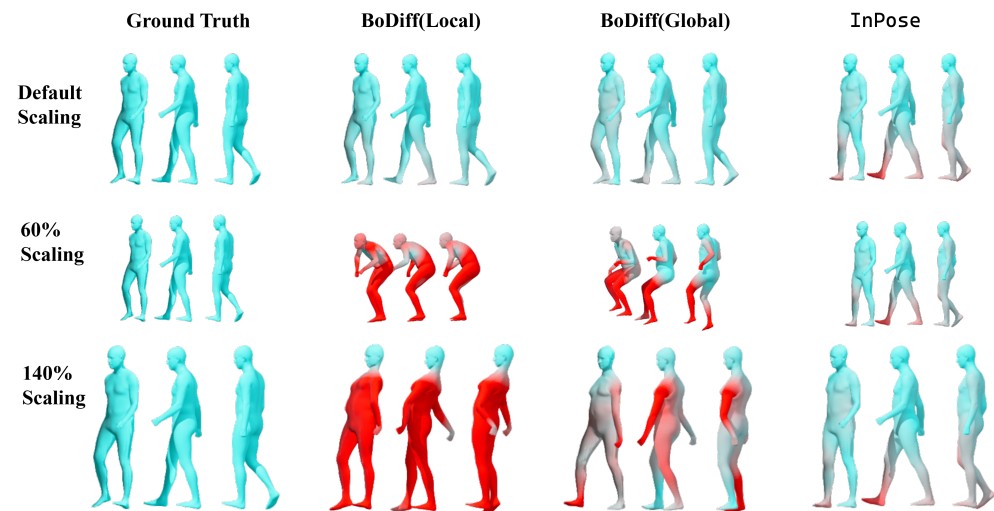

Figure 11: More scaling qualitative results comparing `InPose` with the Diffusion-based Baselines with varying body scale. `InPose` is able to infer lower body movement using the prior learnt from hand motion during walking.

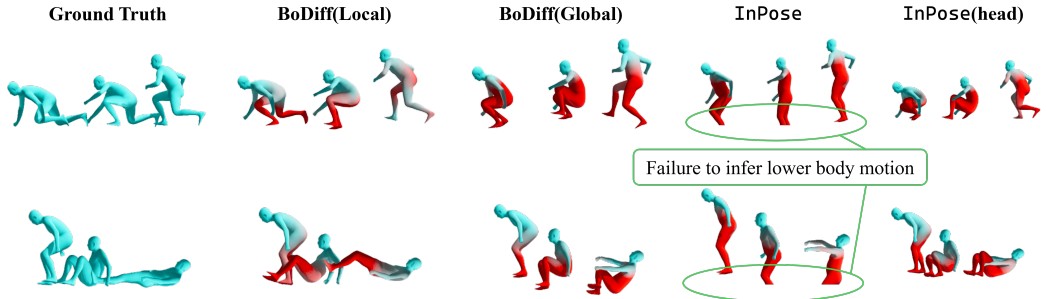

Figure 12: Catastrophic failure cases of `InPose`. This occurs when the user gets extremely close to the ground. Without root translation information, `InPose` fails to infer lower body motion. This is alleviated by using the head position as input for CFG, in `InPose` (head).

