# OpenReview forum: "Zero-shot Human Pose Estimation using Diffusion-based Inverse solvers"
_ICLR.cc/2026/Conference — ICLR 2026 Poster_

### Official Review · Reviewer_ny6y · 2025-10-31

**Soundness:** 1
**Presentation:** 3
**Contribution:** 2
**Rating:** 2
**Confidence:** 4

**Summary:**

The paper proposes InPose, a diffusion-based inverse solver for human pose estimation from sparse sensors. The model assumes access to rotations and positions of only three joints (head and two wrists). It decomposes poses into scale-free and scale-dependent components and uses a pre-trained diffusion to restore full-body poses with sparse observations. During inference, it adopts ideas from IIGDM to incorporate positional likelihood, allowing zero-shot adaptation to new body scales without fine-tuning.

**Strengths:**

1. The paper introduces a diffusion-based inverse inference framework for human pose estimation, which is conceptually original. It demonstrates how diffusion models for forward generation can be adapted to approximate inverse inference by adding an explicit likelihood-guidance term during sampling.
2. The decomposition of human poses into scale-free and scale-dependent components is theoretically motivated.
3. Within the paper’s stated assumptions, the derivation is internally consistent. Although these assumptions are physically unrealistic, the formulation itself is mathematically correct.

**Weaknesses:**

1. The entire formulation assumes that the three sensors directly provide joint rotations r_m. In practice, such rotations cannot be measured without knowing both adjacent body segments’ orientations. This input variable is thus non-observable, and the model seems to operate on dataset-level ground-truth artifacts rather than physically measurable signals.
2. According to Fig.1, the conditioning joints (head and wrists) are leaves in the kinematic tree. While leaf-joint rotations may be defined as parameters in datasets, they are not constraining in the forward-kinematic chain (i.e., not required to determine downstream positions). Conditioning p(r_M|r_m) on such non-constraining variables makes the setup physically non-closed: the learned dependence is statistical rather than geometrically consistent, weakening the realism and deployability of the approach.
3. The paper argues that zero-shot generalization is needed because users have different body scales, yet it simultaneously assumes that the new user’s bone lengths are known at inference time. If scale information is already available, a simpler and physically grounded approach would train a scale-free diffusion prior and condition on the user’s scale-dependent skeleton (e.g. scaled T-pose) together with observable head/wrists measurements (if available and physically meaningful). Such a design would achieve the same finetune-free behavior without introducing complicated conditioning mechanisms.

**Questions:**

1. Could the authors clarify how r_m could be physically obtained from only three sensors? Are these intended as device orientations or joint rotations, or just dataset artifacts?

---

> ### Author Response · Authors · 2025-11-25
>
> We understand the reviewer’s concerns and questions. We may be able to alleviate some of the misunderstandings and confusion (and will accordingly update the paper since we now understand the issues).
>
> * **Formulation assumes that 3 sensors directly provide joint rotations $r_m$. Could the authors clarify how $r_m$ could be physically obtained from only 3 sensors?**
>
>     In our paper, rotation $r_m$ refers to the rotations of the joints in the global coordinate system (not the parent joint’s local coordinate system). For example, if a user is wearing a wristwatch, the wristwatch can estimate its own rotation angle in a global coordinate system. If we know the global joint angles $r_{M}$ of **all joints**, we can use the kinematic tree to compute all the local joint angles as well (which we refer to as $\Theta_{M}$ in the paper).
>
>     One sensor typically used to estimate global rotations $r_m$ is the inertial measurement unit (IMU). When using an IMU, one can fuse measurements from the accelerometer, gyroscope, and magnetometer to establish a global coordinate frame, determined by the direction of gravity and the magnetic North. The gyroscope’s angular velocity measurements can then be projected to this global coordinate frame and integrated over time to compute the rotation angle. Thus, by combining all three IMU sensors, it is possible to track global rotation with high accuracy (especially when the gyroscopes are of high quality, such as those used in military applications). The following papers and products offer details on tracking global orientations:
>     * S. O. H. Madgwick, A. J. L. Harrison and R. Vaidyanathan, "Estimation of IMU and MARG orientation using a gradient descent algorithm," *2011 IEEE International Conference on Rehabilitation Robotics*, Zurich, Switzerland, 2011.
>     * G. Wetzstein, “3-DOF Orientation Tracking with IMUs,” [Online]. Available: https://stanford.edu/class/ee267/notes/ee267_notes_imu.pdf (accessed November 24, 2025)
>
>     We now realize that this may not have been clear in the **Section 2** of our original draft, where we introduced three-point pose estimation; we apologize and will clarify upfront in the revised version.
>
> * **The head and wrist joints are leaves of the kinematic tree and are not constrained in the forward kinematic chain. How can this give geometrically consistent results?**
>
>     We are unsure if we have understood this question correctly. Algorithms that perform sparse sensor-based motion tracking, including ours, perform well *precisely* because they track the leaf joints of the kinematic chain. Since these joints are at the leaf, their position depends on all parent joints leading up to the root joint. We utilize this fact when setting up our kinematic system of equations for inverse guidance **(Equation 8)**. All intermediate joints are included in the system of equations, and their solutions need to be close to ground truth in order for the leaf joints to match the measurements. In other words, if we were to use a joint other than the leaf for setting up our constraints, then the child joints of the chosen constraint joint will be free parameters and won’t be guided by the inverse guidance term (i.e., they would be guided only by the prior).

---

> ### Author Response · Authors · 2025-11-25
>
> * **If bone lengths are known during inference time, then why not train a scale-free diffusion prior and condition on the user’s scale-dependent skeleton?**
>
>     We are not sure whether we fully understand your suggestion. The scale-free observations alone cannot be used to estimate pose, as illustrated in **Table 3**. We would need to condition the algorithm on the scale-dependent joint positions in order to perform accurate pose estimation.
>
>     The conditioning approach that you suggest(augmented with joint positions), where we condition on the user’s skeleton, has been previously used in a few works, as mentioned in our Related Works section **(Section 5)**. One of the works that conditions on user body shape is Aliakbarian et al. (2022) [1]. The authors jointly train sensor inputs and bone length parameters within a flow-based generative model framework. However, this approach requires jointly training pose and a large number of body shapes in order to generalize. In contrast, our work can directly accept any set of body bone lengths, even though it was trained on a single user’s bone lengths. Your suggested approach would be ideal if the deviation in bone lengths is small compared to the default body shape. Such a scenario would be one suggested by Reviewer **tqDn**. In our new **Table 4**, which is also provided in our rebuttal to Reviewer **tqDn,** we perform a comparison when varying bone lengths based on body shapes within the AMASS dataset. The variation studied in **Table 4** is relatively minor compared to the default body shape. This comparison also tests a version of the BoDiffusion baseline where the training data is augmented with the various body shapes within AMASS.
>
>     However, to demonstrate that our algorithm can generalize even across large deviations in body shape, we present comparisons in **Tables 1 and 2**, where we test body shapes substantially different from the default body shape. In all cases, we observe that the MPJRE for InPose remains relatively constant, demonstrating its effectiveness. MPJRE is arguably the most important metric for Pose Estimation, as it is scale-free and independent of the user’s shape.
>
> [1] Sadegh Aliakbarian, Pashmina Cameron, Federica Bogo, Andrew Fitzgibbon, and Thomas J. Cashman. FLAG: Flow-based 3D Avatar Generation from Sparse Observations. In 2022 IEEE/CVF Conference on Computer Vision and Pattern Recognition (CVPR)

---

### Official Review · Reviewer_wWuf · 2025-11-01

**Soundness:** 2
**Presentation:** 3
**Contribution:** 3
**Rating:** 6
**Confidence:** 4

**Summary:**

This paper proposes a new diffusion-based human pose estimation method for sensor data. The core motivation is that human skeletons have diverse limb lengths, so a conditional diffusion model fails to be trained on that diverse data. To bypass this problem, the paper proposes to train the diffusion model only on rotational data, which is independent of limb lengths. To realize this idea, the conditional diffusion formulation is separated into two parts, i.e., rotational distribution and locational distribution. The rotational distribution is absorbed in a classifier-free guidance (CFG) formulation and trained on data, while the location part is simply formulated based on geometry. To handle the location part in reverse steps, the pseudo-inverse guidance technique is utilized. This technique requires the transform to be linear, which is not satisfied by the geometric property of the rotations, so the result of the technique is approximated by a Gaussian distribution. Experiments show that the proposed method is much better generalized for diverse data.

**Strengths:**

- Good design. I'm impressed by the separated CFG design. This cleverly bypasses the difficulty of limb-length differences.

- Good generalization performance.

**Weaknesses:**

- Some crude components/approximations: The approximation for the $\Pi$GDM is somewhat crude. (i) It requires a strong assumption. The assumption in Theorem 1 requires the 6 DoF representation to be already close to a Stiefel matrix. However, this may not be true for a large part of the diffusion process. (ii) Theorem 1 is somewhat misleading. The proof only shows that the mean and the covariance of the distribution will correspond to the derived expressions. This does not imply that the resulting distribution will be close enough to a Gaussian one. It can be a widely different distribution, which still has the same mean and covariance.

- Better generalization, but worse in the best-case scenario? In Figure 3, InPose is worse than existing methods when the scale is one. The paper says that this is "expected," but why is this? I cannot find a valid reason for the proposed formation being worse with the default scale. A more thorough discussion is needed here.

**Questions:**

Please see the above weaknesses.

---

> ### Author Response · Authors · 2025-11-25
>
> We thank the reviewer for their incredibly insightful comments. They brought to our attention a key limitation of our approach.
>
> * **The mismatch between our Gaussian approximation used in Theorem 1 and the true distribution**
>
>     We concede that the assumptions made in Theorem 1 are rudimentary, and that the resulting distribution $p(\mathcal{D}(r^0_M)|r^t_M)$ may not be Gaussian at all, especially because of the nonlinearity $\mathcal{D}()$. However, we also tried other diffusion-based inverse guidance algorithms, including one by Rozet et al. (2024)[1]. This algorithm once again assumes that $p(\mathcal{D}(r^0_M)|r^t_M)$ is a Gaussian, although, once again, the nonlinearity $D()$ may cause a significant deviation from a true Gaussian.
>
>     Rozet et al. (2024) is an extension to $\Pi$GDM, where the Covariance matrix for $p(r^0_M|r^t_M)$  is no longer assumed to be Diagonal. It instead tries to approximate a Covariance fit implicitly, and even works when the nonlinearity $\mathcal{D}()$ is applied to the output $\hat{r}^0_M$ of our diffusion model. We found that it does converge, which indicates that our approximation may not be too far off from a Gaussian. However, it performs slightly worse (about 10% worse error) than our current $\Pi$GDM-based algorithm. We did not report this in the paper because of its increased runtime with worse performance. We suspect that the lack of performance improvement may be due to the imperfect Gaussian approximation, and this is a reason why the algorithm doesn’t beat the baselines in the default body shape scenario.
>
>     We have updated our draft reflecting this comment in the Limitations Section(Appendix D).
>
> * **Why is InPose worse than existing methods (Fig. 3) when the scale is 1?**
>
>     We believe that another reason our method fails to outperform the baselines in the default case, apart from the Gaussian mismatch, is that the problem is fundamentally ill-posed. Although the kinematic model for 3-point pose is linear, the problem is under-determined since we need to infer all 24-joint locations from just three input joint locations. The neural network approaches that serve as our baselines are trained to directly map the three joint locations to the full pose and are able to learn a robust, nonlinear mapping. In contrast, our inverse guidance approach relies on a linear kinematic formulation, which may be less expressive. In fact, using the head joint position as input for CFG improves performance dramatically, as shown in the new **Table 4** for **InPose(Head)**.
>
>     In our next comment, we present results with the default body shape, which includes **InPose(Head)**. As we can see, **InPose(Head)** outperforms **BoDiffusion(Global)**, which serves as the base model on top of which we implement our inverse guidance.
>
> [1]  François Rozet, Gérôme Andry, François Lanusse, and Gilles Louppe. Learning diffusion priors from observations by expectation maximization. In The Thirty-eighth Annual Conference on Neural Information Processing Systems, 2024

---

> ### Author Response · Authors · 2025-11-28
> **Performance in the default body shape case with InPose(head)**
>
> For completeness, here is a performance comparison for **InPose(Head)** where we augment the head position as input for CFG. The test was performed for the **default body shape scenario**, and tested on Protocol 1.
>
> |Algorithm | MPJPE(cm) | MPJRE($\degree$) | UPE(cm) | LPE(cm) |
> | --- | --- | --- | --- | --- |
> | BoDiffusion(Local) |  5.16 | 4.32 | 2.36 | 9.72  |
> | BoDiffusion(Global) | 5.97 | 4.97 | 2.35 | 11.96 |
> | InPose | 7.64 | 6.38 | 3.36  | 14.74 |
> | InPose(Head) | 5.85 | 4.69 | 2.48 | 11.34 |
>
> Here, we see that although InPose(head) is unable to beat BoDiffusion(Local) in the default body shape scenario, it can beat BoDiffusion(Global), which is the base model over which we implement our Inverse Guidance in nearly all metrics.

---

### Official Review · Reviewer_tqDn · 2025-11-02

**Soundness:** 3
**Presentation:** 2
**Contribution:** 2
**Rating:** 4
**Confidence:** 4

**Summary:**

This paper proposes an approach for estimating the 3D human pose from sparse body-mounted sensor measurements (e.g., using three sensors that track position and orientation). The method employs a diffusion-based framework, with the key contribution being the advantage of accommodating different body shapes, unlike prior methods that assume a fixed body shape. The approach is evaluated on the AMASS dataset and compared against previous work.

**Strengths:**

- The proposed method achieves improved robustness compared to previous work, specifically BoDiffusion and AvatarJLM.

**Weaknesses:**

- The evaluation is limited to synthesized data (for noise and different scales).
- The experiment using Gaussian noise is interesting, but this is not necessarily representative of the type of noise that these sensors tend to have.
- The experiments varying the body size introduce some arbitrary scaling of the upper body or the arms/legs. Is an 1.4x scaling for the torso while doing a 0.7x scaling for the arms realistic? Why not sample real motions and real bodies from AMASS for these experiments?
- I understand that the baselines are trained with a specific body shape, but how would they perform if their training data is augmented with arbitrary scaling factors for the different body parts. Is scale robustness something that can be achieved with augmentations?
- Although the setting is different, there has been recent work that operates with arbitrary body shape parameters (EgoAllo, Yi et al, CVPR 2025). How does that compare to the proposed work?
- Minor: The paper refers to the representation of Zhou et al as 6DoF. My understanding is that this representation still has three degrees of freedom, they just regress 6 values, so they refer to it as 6D.

**Questions:**

As I describe in the weaknesses, I would be interested in seeing:
- experiments with more realistic scale variations (e.g., with body shapes from AMASS).
- a comparison (conceptual, and potentially experimental) with the EgoAllo design.
- a version of the baselines where they see data with different scales during training (i.e., augmentations on the body shape).

---

> ### Author Response · Authors · 2025-11-25
>
> We thank the reviewer for their incredibly insightful and stimulative feedback. We hope to answer all their questions in the following rebuttal.
> * **How would the baselines perform if training data is augmented with scaling factors from the different body parts?**
>
>    The AMASS datasets contain motion capture from multiple individuals with some variation in body shapes. This body shape variation was originally ignored in our baselines; however, as you suggested, we fine-tuned **BoDiffusion(local)**(also termed BoDiffusion(L)) with joint positions that were generated from different body shapes in AMASS. This fine-tuning was done in accordance with Protocol 1.
>
>     Note that the scale-free joint rotations of each sample are maintained. In contrast, the scale-dependent joint positions are modified according to the body shape of the user whose motion capture data was used to generate the sample. This fine-tuned baseline is referred to as **BoDiffusion(L)(ft)**. We do not fine-tune the **BoDiffusion(global)** with various body shapes because **BoDiffusion(global)** serves as the base diffusion model for InPose, and we want to demonstrate that InPose can outperform other baselines even when trained solely on the default user's data.
>
>     We also add a slightly modified version of **InPose** for comparison with **BoDiffusion(L)(ft)**. Along with positional inverse guidance, we use the head joint position as an input to the CFG of the diffusion model. When testing body shapes with minor deviations from the default body shape, the height variation across the tested body shapes is also minor. Thus, the head joint position should be similar as well. This modified version is termed **InPose(head)**.
>
>     We report results with these two modified algorithms in the following section:
> * **Is 1.4x scaling for the torso and 0.7x scaling for the arms realistic? Why not sample from the AMASS dataset?**
>
>     The body size variations in the AMASS dataset are limited. Since we aim to demonstrate the generality of InPose, we selected scaling factors in Tables 1 and 2 that correspond to the outliers of the body-shape distribution. If one wanted to solve only for the typical body shapes, then **Reviewer n6y6’s** suggestion to train a model—conditioned on slight variations in bone lengths—would suffice. However, with our inverse guidance approach, we are not limited by the sets of bone lengths used during training; we can generalise to any body shape.
>
>     Regardless, as suggested, we present new test results by varying body shape according to the users in AMASS, as this may be more “typical” in real-world scenarios. We selected the upper body shapes in this ablation after determining the variation in bone lengths among users in AMASS. To determine the upper body variation, we first calculated the ratio of torso length $d_{torso}$ to height $d_{height}$ for each user $\gamma = \frac{d_{height}}{d_{torso}}$. Since we want transferability between users, we want a model trained on one user to be capable of inferring on all other users. For this reason, we choose the upper body multiple to be the ratio between the minimum $\gamma_{min}$ and maximum $\gamma_{max}$ across the users, which we found to be about 0.85($=\frac{2.28}{2.68}$). We also use the inverse of this ratio (1.17). For the arm length variation, we perform a similar computation and found it to be 0.85 as well. The results are presented in **Table 4** of the paper, and we summarize the primary outcome here.
>
>     We provide the table for the **(Arms x0.85, torso x1.17)** case below; results for the other cases tested are provided in **Table 4**. In this comparison, we find that although the MPJRE of **InPose** without head input stays consistent across body shapes, it falls behind the other baselines when tested with relatively minor variation in bone lengths that we computed above from AMASS. However, by simply adding the head position as CFG input to the diffusion model, **InPose(head)** can outperform all the baselines for the tested variation in body shapes in all metrics except Lower Body Position Error(LPE). In the other comparisons, (Torso x0.85) and (Torso x1.17), **InPose(head)** outperforms all other algorithms in **all** metrics.
>
>     Secondly, it appears that augmenting the training dataset for the baselines with various body shapes from AMASS for **BoDiffusion(L)(ft)** doesn’t allow it to comprehensively trounce training with the default body shape (**BoDiffusion(Local)**), as seen in the MPJRE and LPE in the (Arms x0.85, torso x1.17) case.
> |Algorithm | MPJPE(cm) | MPJRE($\degree$) | UPE(cm) | LPE(cm) |
> | --- | --- | --- | --- | --- |
> | BoDiffusion(Local) |   6.27 | 4.91  |  3.64 | **10.58** |
> | BoDiffusion(Glocal) |   6.8 | 5.7  | 3.36 | 12.49 |
> | BoDiffusion(L)(ft) | 6.17 | 5.06  | 3.22  | 11.04 |
> | InPose | 7.8  | 6.33  | 3.46  | 14.95 |
> | InPose(Head) | **5.71**  | **4.5**  | **2.37** | 11.16 |

---

> ### Author Response · Authors · 2025-11-25
>
> * **How does this paper compare against recent work, EgoAllo(Yi et al, CVPR 2025 [1])?**
>
>     Thanks for this pointer to a very recent work. We spent some time on implementing this; here is the summary of our conceptual/experimental comparison.
>
>     **EgoAllo**’s inputs are positions and rotations of the head, similar to ours, but their hand/finger poses come from an egocentric camera. We can formulate their problem as having head position $l_{head}$ and head rotation $r_{head}$ as inputs, along with *intermittent* wrist positions and rotations $l_{wrists}$ and $r_{wrists}$. EgoAllo proposes an invariant representation for diffusion model inputs, but they utilize a slightly modified head position input into their diffusion model as CFG.
>
>     Instead of using a $\Pi$GDM inverse guidance term, EgoAllo uses a Nonlinear Least Squares (NLLS) optimizer (Levenberg-Marquardt) to guide their model by directly optimizing the model's outputs (joint angles) at every diffusion step. It does not compute any gradients through the diffusion model. We believe this is suboptimal because the inverse guidance updates aren't regularized by the prior learnt by the diffusion model.
>
>     To compare EgoAllo’s guidance framework with ours, we first tested their algorithm with a slight modification to support 3-point pose inference on the AMASS dataset with Protocol 1. In their original work, the authors use the output from a vision-based depth algorithm to intermittently provide the wrist locations and orientations. We provide these inputs continuously using the ground-truth data. This is termed **EgoAllo(Mod)**.
>
>     We also implemented their guidance framework within BoDiffusion, where we substituted our $\Pi$GDM-based inverse guidance with their NLLS optimizer at the output of each diffusion step. We term this **BoDiffusion(Ego)**. We needed to slightly modify the optimization objective described in their paper for **BoDiffusion(Ego)**, as we do not have access to all the inputs and outputs specified in their setup. Specifically, we don't use their reprojection loss since we do not have camera inputs, nor do we use their skate loss since BoDiffusion does not provide contact point estimates (contact points are also infeasible for many of our testing samples, as many of them involve users jumping/sleeping). We use their prior loss, velocity smoothing loss, and their 3D reconstruction loss while testing, each of which is scaled by the same hyperparameters reported in their paper.
>
>     The results are summarized in the **table** below. We can clearly see that both InPose and InPose(head) outperform EgoAllo(Mod) in both MPJPE and MPJRE. The regular InPose is able to beat EgoAllo(Mod) despite not using the head position as input for CFG. (The *MPJRE* for EgoAllo(Mod), which was calculated using our own script, seems a little high in relation to its *MPJPE*, which was calculated by the authors' script.)
>
>     In our testing, we found that using EgoAllo’s NLLS guidance does not make the **BoDiffusion(Ego)** algorithm converge when positional joint inputs are not provided as CFG. More concretely, we observe that because the earlier diffusion step outputs are highly noisy, the NLLS optimizer converges to a local optimum that does not belong to the prior distribution of human poses. This results in the diffusion model converging to an inaccurate pose sequence after 50 steps. We hypothesize that EgoAllo’s guidance converges in their tests but not ours because their preliminary Diffusion model may be more expressive than BoDiffusion, and that the skating losses, which we skip, are critical.
>
>     That said, we did find that their NLLS optimizer can improve performance metrics in certain test samples when used for post-processing. We first use a different algorithm (BoDiffusion or InPose) to obtain an initial estimate of the user’s pose, and then utilize EgoAllo’s NLLS guidance to optimize the final pose sequence. This provided an improvement of about $\sim 5$% in the favourable samples.
>
> |Algorithm | MPJPE(cm) | MPJRE($\degree$) |
> | --- | --- | --- |
> | EgoAllo(Mod) | 7.67 | *11.72* |
> | BoDiffusion(Ego) |  - | - |
> | InPose | 7.64 | 6.38 |
> | InPose(Head) | **5.85** | **4.69** |
>
> [1]  Brent Yi, Vickie Ye, Maya Zheng, Yunqi Li, Lea Müller, Georgios Pavlakos, Yi Ma, Jitendra Malik, and Angjoo Kanazawa. Estimating body and hand motion in an ego-sensed world.
> CVPR 2025

---

> ### Author Response · Authors · 2025-11-25
>
> * **(Minor) Regarding the 6D terminology used throughout the paper:**
>
>     You are correct, we use the term 6DoF even though the representation itself has only 3 degrees of freedom. We will ensure that we substitute '6DoF' with '6D’ in our updated draft.
>
> * **Changes to the draft**
>
>     We have updated the paper draft in response to all the reviewers' comments. The added sections are mostly at the end of the Appendix, and are highlighted in yellow.

---

### Official Review · Reviewer_zAJP · 2025-11-03

**Soundness:** 3
**Presentation:** 3
**Contribution:** 3
**Rating:** 8
**Confidence:** 4

**Summary:**

This paper introduces InPose, a diffusion-based method for human pose estimation from sparse sensor data (3 sensors: head and two wrists). The key innovation is formulating pose estimation as an inverse problem, enabling zero-shot generalization across users with different body sizes—without requiring model fine-tuning for each user. InPose leverages a pre-trained diffusion model conditioned on rotational measurements, using location measurements only as a likelihood-based guidance term during inference. The method is evaluated on the AMASS dataset and compared to state-of-the-art baselines, showing strong generalization and robustness to body size and measurement noise.

**Strengths:**

+ Novel Inverse Problem Formulation: The decomposition of human pose into scale-free (rotational) and scale-dependent (location) components is elegant and addresses the generalization issue in prior work.
+ Zero-Shot Generalization: InPose can handle unseen body sizes and shapes without retraining, a significant practical advantage.
+ Robustness: The method is robust to measurement noise, as shown in experiments with noisy sensor data.
+ The paper provides mathematical justification for the Gaussian approximation in the likelihood term and details the propagation of uncertainty through non-linear operators.
+ Evaluations cover generalization to new body sizes, robustness to noise, and ablation studies on representation choices (6DoF vs. rotation matrices).

**Weaknesses:**

- Performance on Default Body Size: For users matching the training body size, baseline methods outperform InPose.
- Complexity: The method involves non-trivial mathematical machinery (e.g., modified ΠGDM, covariance propagation), which may hinder adoption. Please include complexity analysis.
- The proposed method may be hard to be used in online real-time applications, where most of the real usages cases in VR and AR require.
- The proposed method builds on top of several previous methods, reused and simplified some formulations. Please clearly define the novelty over these previous methods.

**Questions:**

Please address the concerns in the weakness session.

---

> ### Author Response · Authors · 2025-11-25
>
> We thank the reviewer for identifying the core contributions in this paper and for asking thoughtful questions.
>
> * **Why do baselines outperform InPose for the default body shape?**
>
>     (1) Observe that full-body pose estimation is a highly ill-posed problem (we need to estimate 24 joint locations using only 3 joint measurements). In this setting, although the true kinematic model for the 3-point measurement is linear, using this linear model for inverse guidance may not be sufficiently expressive to fully mitigate the ill-posed nature of the problem. In contrast, a neural network trained on the default body shape can learn a powerful non-linear mapping from the 3-joint locations to the full 24-joint body pose. Once the body shape changes, the non-linear mapping that is overfitted to the default body shape begins to fail. In contrast, our InPose approach remains stable because it models the body's kinematics in its inverse guidance.
>
>     We observe that simply augmenting **InPose** by using the head position as the CFG input (termed **InPose(Head)**) enables us to outperform the base **BoDiffusion(Global)** model, on which we implement our inverse guidance, even for the default body shape. These results are shown in our third response to this review.
>
>     (2) A second reason could be the Gaussian approximations made in Theorem 1, which inject some inaccuracies into our inverse guidance model. But a neural network, trained and tested on the same body shape, absorbs (or overfits) to all unmodeled parameters.
>
> * **Complexity of the mathematical machinery in the method?**
>
>     Our approach uses BoDiffusion (Castillo et al. (2023) [2]) as the backbone Diffusion model. We adapt the same Diffusion Transformer architecture that they use, which uses 22M parameters. During inference, we perform 50 diffusion steps. Since we use $\Pi$GDM for inverse guidance, at each diffusion step, we perform one forward neural network invocation, as well as one backward invocation to obtain the required gradients. At the end of 50 diffusion steps, we are left with 41 samples, although we use a 20-sample overlap to ensure continuity. We describe runtime performance in the following paragraph. We will also upload our code, along with all necessary requirements and documentation, to facilitate further progress in the field.
>
> * **Can this be used in real-time for AR/VR applications?**
>
>     We concede that InPose is not ready to run in real-time on current embedded AR/VR computing hardware, such as glasses and goggles. However, we think InPose can still be helpful in scenarios where the user is connected to a separate computing unit, such as a desktop. Our algorithm is capable of running in real-time on an RTX Titan, a 7-year-old GPU, albeit with some latency. We can generate about 229 samples per second, which, at 60 frames per second, results in us needing approximately 0.25 seconds for every second of pose sequences. If necessary, we can also trade off performance for runtime by reducing the number of diffusion steps — as shown in a new result in **Figure 11**. We provide runtime comparisons with the baselines in a new **Table 5**, which we also report below:
> | Algorithm | Samples/sec |
> | --- | --- |
> | AvatarJLM | $\sim$102 |
> | BoDiffusion | $\sim$392 |
> | InPose | $\sim$229 |

---

> ### Author Response · Authors · 2025-11-25
>
> * **Clarify novelty from past methods?**
>
>     Our novelty is rooted in (1) formulating full-body pose-estimation as an inverse problem, and (2) decomposing the pose estimation problem into a scale-free and scale-dependent component, which allows us to plug the problem into the inverse-problem framework. Once this formulation has been developed, we utilize the existing $\Pi$GDM algorithm to execute the internal inverse guidance. However, $\Pi$GDM cannot be applied blindly, given the nature of our kinematics operator and the 6D representation of joint rotations. Hence, we first show Gaussian approximations to establish the validity of $\Pi$GDM and then utilize $\Pi$GDM to solve the problem.
>
>     In contrast, SOTA baselines attack the pose estimation problem from two main directions. The first approach trains diffusion models on datasets gathered from a relatively small variety of body shapes available in the AMASS datasets, but this does not generalize to atypical body shapes. The second approach, e.g., **EgoAllo** [1] from CVPR 2025, introduces a separate guidance term to make their neural network-based algorithm scale-free. However, to the best of our knowledge, ours is the first work to perform **principled** inverse guidance for general diffusion models. Unlike EgoAllo, whose guidance is formulated as a nonlinear optimization problem that isn’t guaranteed to converge, our inverse guidance is regularized by the prior learned by the diffusion model itself. This enables our approach to work on a broader range of diffusion architectures. Empirically, we find that EgoAllo fails to outperform InPose on the AMASS dataset for 3-point pose estimation. The guidance term introduced in EgoAllo is also incompatible with our base Diffusion model and doesn’t converge. We have provided further details in our response to **Reviewer tqDn**.
>
> [1]  Brent Yi, Vickie Ye, Maya Zheng, Yunqi Li, Lea Müller, Georgios Pavlakos, Yi Ma, Jitendra Malik, and Angjoo Kanazawa. Estimating body and hand motion in an ego-sensed world.
> CVPR 2025
>
> [2] Angela Castillo, Maria Escobar, Guillaume Jeanneret, Albert Pumarola, Pablo Arbeláez, Ali Thabet, and Artsiom Sanakoyeu. Bodiffusion: Diffusing sparse observations for full-body human motion synthesis. In Proceedings of the IEEE/CVF International Conference on Computer Vision, 2023

---

> ### Author Response · Authors · 2025-11-28
> **Performance in the default body shape case with InPose(head)**
>
> For completeness, here is a performance comparison for **InPose(Head)** where we augment the head joint position input for CFG. The wrist input is performed through inverse guidance itself. The test was performed for the **default body shape scenario**, and tested on Protocol 1.
>
> |Algorithm | MPJPE(cm) | MPJRE($\degree$) | UPE(cm) | LPE(cm) |
> | --- | --- | --- | --- | --- |
> | BoDiffusion(Local) |  5.16 | 4.32 | 2.36 | 9.72  |
> | BoDiffusion(Global) | 5.97 | 4.97 | 2.35 | 11.96 |
> | InPose | 7.64 | 6.38 | 3.36  | 14.74 |
> | InPose(Head) | 5.85 | 4.69 | 2.48 | 11.34 |
>
> Here, we see that although InPose(head) is unable to beat BoDiffusion(Local) in the default body shape scenario, it can beat BoDiffusion(Global), which is the base model over which we implement our Inverse Guidance in nearly all metrics.

---

### Author Response · Authors · 2025-11-27

We thank all the reviewers for their insightful comments. We hope that we have clarified all your questions and concerns about this paper. We would be extremely grateful if you could provide remarks on our rebuttal as soon as possible, so that we may respond to them before the end of the discussion period on **December 2nd**.

We have added **two additional tables** and **one additional figure** to our draft at the end of the Appendix, and have highlighted all the added text. The added tables incorporate a slightly modified version of InPose as well as a modified baseline, both of which we have described in our response to **Reviewer tqDn**. We have also incorporated some of the suggested changes to the rest of the paper.

---

### Meta-Review · Area_Chair_6wUP · 2025-12-07

**Summary:**

The paper presents a diffusion-based method for zero-shot human pose estimation, leveraging sparse sensor data (head and wrists) to estimate full-body poses. The key innovation of the paper lies in decomposing the pose into scale-free (rotational) and scale-dependent (location) components, thereby enabling the model to generalize across different body sizes without retraining. While the method demonstrates strong generalization and robustness, several reviewers have raised concerns about specific aspects of the approach, such as the performance on default body shapes, the complexity of the mathematical framework, and the physical realism of certain assumptions, particularly in terms of obtaining rotation data from just three sensors. These issues have been addressed in the rebuttal, with clarifications and additional experiments presented to support the proposed method's validity.

**Reviewer Concerns:**

Addressed Concerns:

Generalization Across Body Shapes: The reviewers raised concerns about the method's performance on default body shapes compared to the baselines. The authors clarified this by presenting results for a modified version of their approach (InPose(Head)) that outperforms the baseline in several scenarios, including default body shapes.

Complexity of the Approach: The complexity of the method, especially with the mathematical machinery involved, was acknowledged. The authors provided more details on the runtime performance and trade-offs that can be made to balance accuracy and computational efficiency.

Physical Feasibility of Rotation Measurement: One reviewer questioned the physical feasibility of obtaining joint rotations with just three sensors. The authors clarified that global joint rotations can be accurately measured using IMUs (Inertial Measurement Units), which are common in sensor-based motion tracking systems.

Outstanding Concerns:

Gaussian Approximation: A reviewer expressed concerns regarding the Gaussian approximation in the likelihood term, questioning its accuracy. The authors have acknowledged this limitation and provided additional discussion, but it remains a point of contention.

Performance in Real-World Settings: The method's potential for real-time applications, especially in AR/VR scenarios, was questioned. The authors addressed this by acknowledging that the method is not currently suitable for embedded AR/VR systems but is feasible with external computing units.

Comparison to EgoAllo: One reviewer requested a more thorough comparison with recent work like EgoAllo (CVPR 2025). The authors responded with a detailed experimental comparison, demonstrating that InPose outperforms EgoAllo in specific scenarios, though EgoAllo’s method could improve performance in certain cases when used as post-processing.

**Reviewer Scores:**

Reviewer zAJP (Hao Jiang):

Rating: 8 (Accept, good paper, poster)

Changes: Reviewer would likely remain confident in their evaluation after the rebuttal. They acknowledged the robustness of the method but noted the challenges in the default body shape case. Their score would likely stay the same, recommending a poster acceptance.

Reviewer tqDn (Georgios Pavlakos):

Rating: 4 (Marginally below the acceptance threshold, but would not mind if accepted)

Changes: Reviewer might adjust their score upward in terms of soundness, given the additional clarifications on performance with the head position input. However, concerns about physical realism may still impact their overall rating, leaning toward acceptance but with some reservations.

Reviewer wWuf (Minsik Lee):

Rating: 6 (Marginally above the acceptance threshold, but would not mind if rejected)

Changes: After the rebuttal, the reviewer may slightly increase their rating for contribution based on the clarification of the Gaussian approximation and other contributions. However, issues with the theoretical assumptions and the complexity of the method still detract from their soundness score.

Reviewer ny6y (Yucheng Xing):

Rating: 2 (Reject, not good enough)

Changes: Despite the authors' rebuttal, the reviewer likely maintains their rejection recommendation due to concerns over the physical feasibility of rotation measurements and the unrealistic assumptions in the method. The rebuttal did not fully address these core concerns for this reviewer.

---

### Decision · Program_Chairs · 2026-01-26

Accept (Poster)